# External trigeminal nerve stimulation in youth with ADHD: a randomized, sham-controlled, phase 2b trial

Aldo Alberto Conti [1,2,19], Natali Bozhilova[1,19], Irem Ece Eraydin [3,4,19], Dominic Stringer [5,6], Lena Johansson [1], Robert Marhenke [3], Andrea Bilbow[7], Sahid El Masri [1,8], Joshua Hyde[3], Giovanni Giaroli[9], Holan Liang [10,11], Federico Fiori[1,12,13], Mitul Ashok Mehta [14], Paramala Santosh [1,12,13], Ben Carter [5,6], Samuele Cortese [3,15,16,17,18] & Katya Rubia [1,8] ✉

External trigeminal nerve stimulation (TNS) received US Food and Drug Administration clearance in 2019 as the first device-based, non-pharmacological treatment for attention-deficit/hyperactivity disorder (ADHD), based on a small pilot sham-controlled randomized controlled trial (RCT) that reported symptom improvement in 62 children with ADHD. Here we conducted a confirmatory multicenter, double-blind, randomized, sham-controlled, parallel-group, phase 2b RCT to investigate short-term and long-term efficacy (6 months) of real versus sham TNS in 150 children and adolescents with ADHD. Participants were randomized to receive real TNS ($n$ = 75, mean age (s.d.) = 12.6 (2.8) years) or sham TNS ($n$ = 75, mean age (s.d.) = 12.6 (2.8) years) nightly for approximately 9 hours for 4 weeks. Bilateral stimulation targeted V1 trigeminal branches using battery-powered electrodes applied to the forehead. Sham TNS delivered 30 seconds of stimulation per hour at lower frequency and pulse width. Intention-to-treat analysis showed no significant differential treatment effects on ADHD symptoms (primary outcome) (estimated adjusted mean difference = 0.83; 95% confidence interval: −2.47 to 4.13; $P$ = 0.622; Cohen's $d$ = 0.09). No serious adverse events were reported, and side effects did not differ between groups. In conclusion, TNS is a safe intervention but does not demonstrate clinical efficacy for pediatric ADHD. Trial registration: ISRCTN82129325.

ADHD is the most common neurodevelopmental condition with a prevalence of approximately 5% in school-age children[1] and is defined by symptoms of inattention and/or hyperactivity/impulsivity that are inconsistent with the developmental stage and substantially impair daily functioning[2]. ADHD is also associated with impairments in executive functions, including in tasks of sustained attention and vigilance[3,4]. They furthermore have small but consistent differences in functional and structural brain regions and networks, most prominently involving frontal, striato-thalamic, parieto-temporal and cerebellar regions[5,6].

Stimulant medications (including methylphenidate and amphetamine) are first-line treatments for severe ADHD, improving symptoms in approximately 70% of children, with effect sizes of about 0.8–1.0 in the short term[7]. However, stimulants can cause side effects, may not be indicated with some associated conditions such as cardiovascular disorders[7], and adherence over time is poor, particularly in adolescence[8]. Furthermore, their longer-term efficacy has not been demonstrated[9], with imaging studies suggesting brain adaptation[10] and, hence, possibly reduced effects with long-term use. Non-stimulants

(for example, atomoxetine, guanfacine or clonidine), considered second-line medications, have on average lower efficacy than stimulants and can also lead to intolerable side effects[7]. Both stimulants and non-stimulants have shown to also improve performance in executive function tasks, including sustained attention and vigilance in children and adults with ADHD[11]. However, importantly, users and their families prefer non-pharmacological treatments with better side effect profiles[12]. However, evidence for the efficacy of interventions such as behavioral therapy, cognitive and parent training, dietary changes and neurofeedback in improving ADHD symptoms remains limited[6,13].

External TNS was granted clearance by the US Food and Drug Administration (FDA) in 2019 as the first non-pharmacological treatment for ADHD. TNS is a non-invasive brain stimulation technique that targets the supratrochlear and supraorbital branches of the ophthalmic division (V1) of the trigeminal nerve by delivering an electric current through electrodes placed on the forehead. Sensory inputs from the trigeminal nerve fibers activate the locus coeruleus, raphe nuclei and nucleus tractus solitarius that innervate in a bottom-up manner several other brain regions, most prominently thalamic, frontal and limbic regions[14,15], all of which are affected in ADHD[5,6]. The effects of TNS on the locus coeruleus and brainstem are thought to enhance attention and arousal mechanisms[15,16], which are commonly affected in ADHD[3,4,17]. Furthermore, TNS is thought to stimulate the release of neurotransmitters important for arousal, attention and emotion regulation, particularly noradrenaline, but also dopamine, glutamate, gamma-aminobutyric acid and serotonin[14], all of which have been implicated in ADHD[18]. Our recent meta-analysis showed that TNS is safe with good tolerability for neurological and psychiatric conditions[19].

The evidence for FDA clearance was based on a pilot double-blind RCT in 62 unmedicated children[20], showing that 4 weeks of nightly real versus sham TNS significantly decreased parent-rated ADHD symptoms on the ADHD Rating Scale (ADHD-RS)[21], with medium effect size (Cohen's $d$ = 0.5). The behavioral effects were correlated with increased electroencephalography activity in right inferior/dorsolateral prefrontal cortex[20], a key region known to be underactive in ADHD[5,22–25]. TNS was well tolerated, with no severe adverse events and only minor, transient side effects, predominantly headaches and fatigue[20,26].

These promising findings call for replication in a definitive, multicenter trial. Furthermore, the pilot study did not assess effects beyond 4 weeks and was limited to very young children aged 8–12 years[20]. To address this need, we conducted a confirmatory, multicenter, double-blind, randomized, sham-controlled, parallel-group, phase 2b trial investigating both short-term (4 weeks) and longer-term (6 months) efficacy of real versus sham TNS not only in children but also in adolescents with ADHD, a population with particularly high need for alternative treatments due to low medication adherence rates[8].

We hypothesized that 4 weeks of nightly real versus sham TNS in children and adolescents with ADHD would improve core symptoms, as measured by parent-rated scales (primary outcome). Secondary cognitive and clinical outcomes included behavioral features associated with ADHD, such as symptoms of depression and anxiety, emotional dysregulation, mind-wandering and sleep as well as performance in a vigilance task. There is consistent evidence that children with ADHD have increased mind-wandering, which interferes with their cognitive performance, in particular in tasks of sustained attention and vigilance[27]. We also used objective measures to investigate the effects of TNS on arousal via pupillometry and on objective hyperactivity using a wrist-worn device. Mechanisms of action were explored through functional magnetic resonance imaging (fMRI), which will be reported separately.

## Results

### Participant disposition
Participants were recruited from September 2022 to November 2024. Data collection including follow-up ended in March 2025.

Of 843 children/adolescents with ADHD and their parents/carers who were interested in the study, 165 provided written informed consent, and 150 (97 males, 64.7%) were enrolled in the study and included in the intention-to-treat (ITT) analysis (Fig. 1). Participants had a mean age (s.d.) of 12.6 years (2.8), and most were of White ethnicity ($n$ = 119, 79.3%) and off medication/medication-naive ($n$ = 91, 60.7%) (Table 1). Although the inclusion criterion for the age range was 8–18 years at the consent stage, four children turned 19 before randomization took place. At baseline, 39.3% of participants were on stable stimulant medication (stimulant medication type, mean dose and dose ranges are reported in Extended Data Table 1); 12.6% were taking other psychotropic medication; and 13.3% were receiving other types of medication (for further demographic and medication information, see Supplementary Table 1 and Extended Data Table 2). All participants met criteria for a Diagnostic and Statistical Manual of Mental Disorders, Fifth Edition (DSM-5)[2] ADHD diagnosis. Among them, 133 participants (88.7%) met criteria for ADHD combined presentation, 16 (10.7%) for ADHD inattentive presentation and one (0.7%) for ADHD hyperactive/impulsive presentation. Comorbid oppositional defiant disorder was present in 54 participants (36%), and conduct disorder was present in four participants (2.7%) (Table 1). At baseline, participants had a mean (s.d.) ADHD-RS total score of 35.3 (9.75), indicating severe ADHD symptomatology (Table 2).

Participants were randomly allocated to real TNS ($n$ = 75, mean age (s.d.) = 12.6 (2.8) years, off medication/medication-naive ($n$ = 46, 61.3%)) or sham TNS ($n$ = 75, mean age (s.d.) = 12.6 (2.8) years), off medication/medication-naive ($n$ = 45, 60%). One hundred and forty (93.3%) participants adhered to the intervention, with only nine participants (real TNS ($n$ = 6, (8%)); sham TNS ($n$ = 3, (4%))) discontinuing the intervention permanently prior to the week 4 primary endpoint (Fig. 1 and Supplementary Table 2). Ten participants (real TNS ($n$ = 7, (8%)); sham TNS ($n$ = 3, (4%))) did not meet the predefined adherence threshold (≥1 hour of device use per night on at least 17 nights), as specified in the statistical analysis plan found in the protocol supplementary material[28]. This includes two participants for whom adherence data were missing, as the sleep diary was not returned (both in the real TNS group). Two participants (one in each group) who said they permanently discontinued the intervention did nevertheless meet the definition for adherence to the intervention. One participant who stated that they completed the intervention (in the sham group) did not meet the adherence threshold.

### Blinding
At the end of week 1 and week 4 of the TNS treatment period, children/adolescents, parents/carers and researchers were asked to guess treatment allocation. Blinding appeared successful at week 1, with high rates of 'don't know' responses across children (40%), parents (50%) and researchers (75.3%). Among children in the real TNS group, 45.3% guessed that they were receiving the real treatment and 12% guessed sham. In the sham TNS group, 45.3% guessed real and 17.3% guessed sham. For parents in the real TNS group, 40% thought that their child was receiving the real treatment and 16% guessed sham; in the sham group, 21.3% guessed real and 22.7% guessed sham. For researchers in the real TNS group, 10.7% guessed real and 10.7% guessed sham; in the sham group, 12.0% guessed real and 16% guessed sham.

Blinding remained successful for most participants at week 4, with 'don't know' responses still reported by 34.9% of children, 33.6% of parents and 53.0% of researchers. Among those who did guess at week 4, guesses were balanced across treatment groups: for children in the real TNS group, 37.8% guessed real and 28.4% guessed sham; for children in the sham TNS group, 32% guessed real and 29.3% guessed sham. For parents in the real TNS group, 37.8% guessed real and 32.4% guessed sham; for parents in the sham TNS group, 21.3% guessed real and 38.7% guessed sham. For researchers in the real TNS group, 13.5% guessed real and 28.4% guessed sham; in the sham group, 13.3% guessed real and 36.0% guessed sham (for further details, see Extended Data Table 3).

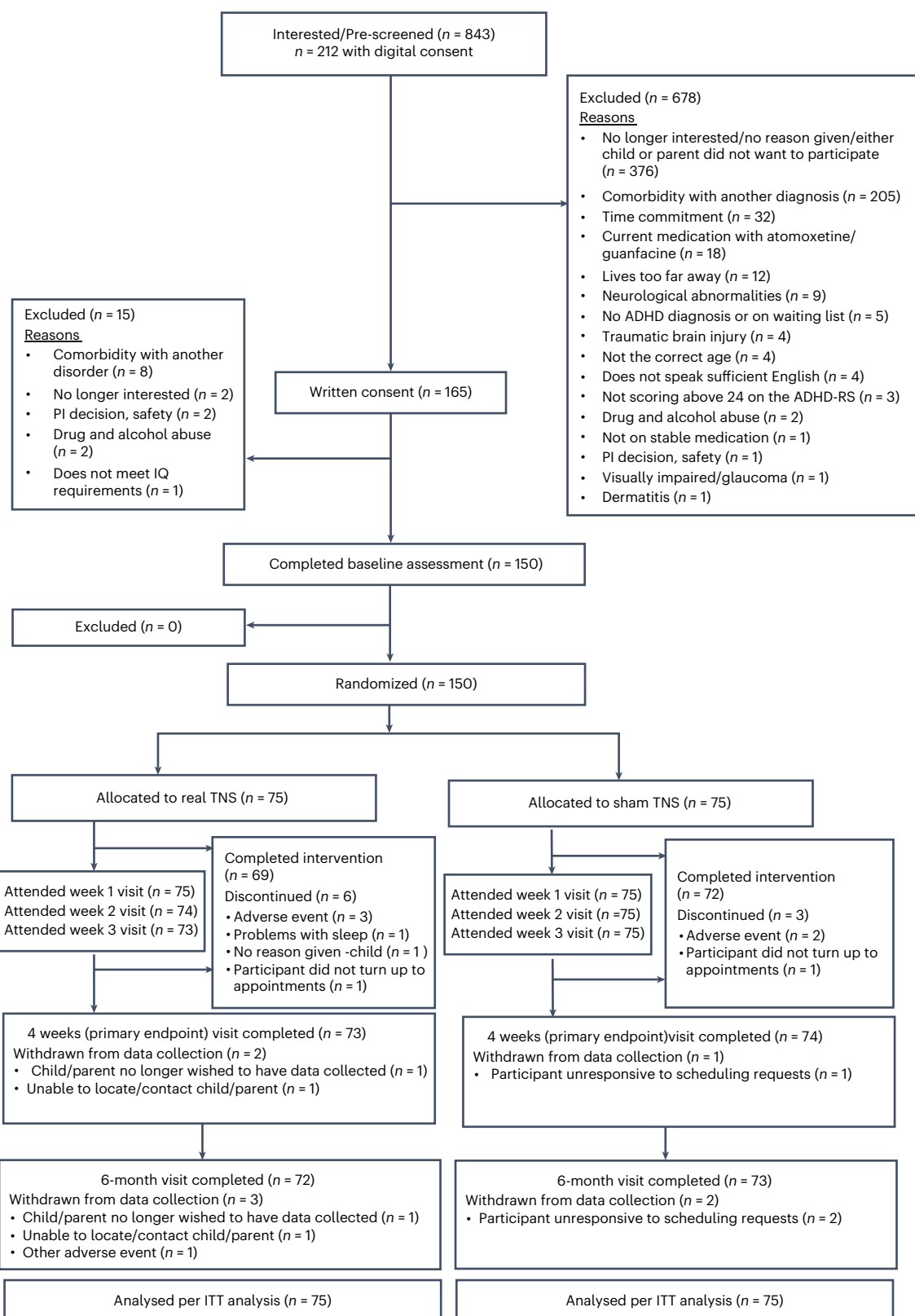

**Fig. 1 | CONSORT diagram.** CONSORT flow diagram of the progress through the phases of enrollment, intervention allocation, follow-up and data analysis for the two treatment arms in the randomized sham-controlled, phase 2b trial testing the efficacy of external TNS in youth with ADHD.

## Primary outcome

ADHD-RS total scores decreased in both groups over the 4-week treatment period, followed by a slight increase from week 3 to week 4 (Fig. 2 and Table 2). At the week 4 primary endpoint, no significant difference was observed between groups (estimated adjusted mean difference (aMD) = 0.83; 95% confidence interval: −2.47 to 4.13; $P$ = 0.622; Cohen's $d$ = 0.09), indicating no evidence of a differential treatment effect between groups (Fig. 2 and Table 2).

## Table 1 | Baseline demographics and clinical characteristics

| Baseline characteristics (n, %) | Real TNS (n=75) | Sham TNS (n=75) | Overall (n=150) |
|---|---|---|---|
| Age (mean, s.d.) | 12.6 (2.8) | 12.6 (2.8) | 12.6 (2.8) |
| Age (categorized per randomization stratifier) | | | |
| 8–13.5 years | 43 (57.3) | 43 (57.3) | 86 (57.3) |
| 13.6–19 years | 32 (42.7) | 32 (42.7) | 64 (42.7) |
| Child sex at birth | | | |
| Male | 49 (65.3) | 48 (64.0) | 97 (64.7) |
| Female | 26 (34.7) | 27 (36.0) | 53 (35.3) |
| ADHD diagnosis per K-SADS | | | |
| Combined presentation | 66 (88.0) | 67 (89.3) | 133 (88.7) |
| Inattentive presentation | 8 (10.7) | 8 (10.7) | 16 (10.7) |
| Hyperactive/impulsive presentation | 1 (1.3) | 0 (0.0) | 1 (0.7) |
| Oppositional disorder per K-SADS | 26 (34.7) | 28 (37.3) | 54 (36.0) |
| Conduct disorder per K-SADS | 4 (5.3) | 0 (0.0) | 4 (2.7) |
| Current stimulant medication status | | | |
| On stable medication | 29 (38.7) | 30 (40.0) | 59 (39.3) |
| Off medication/naive | 46 (61.3) | 45 (60.0) | 91 (60.7) |
| WASI FSIQ-4 score (mean (s.d.)) | 105.5 (13.8) | 109.8 (13.5) | 107.6 (13.8) |
| Child ethnicity | | | |
| White | 61 (81.3) | 58 (77.3) | 119 (79.3) |
| Black, African, Caribbean or Black British | 4 (5.3) | 1 (1.3) | 5 (3.3) |
| Asian or Asian British | 2 (2.7) | 5 (6.7) | 7 (4.7) |
| Mixed or multiple ethnic groups | 6 (8.0) | 9 (12.0) | 15 (10.0) |
| Other ethnic groups | 2 (2.7) | 2 (2.7) | 4 (2.7) |
| Handedness | | | |
| Right handed | 60 (80.0) | 52 (69.3) | 112 (74.7) |
| Left/mixed handed | 15 (20) | 23 (30.6) | 38 (50.6) |
| Index of Multiple Deprivation (mean (s.d.)) | 6.5 (2.8) | 6.8 (2.6) | 6.7 (2.7) |
| Site | | | |
| King's College London | 54 (72.0) | 57 (76.0) | 111 (74.0) |
| University of Southampton | 21 (28.0) | 18 (24.0) | 39 (26.0) |

Categorical variables are presented as the number of participants, with the percentage in parentheses. Continuous variables are reported as mean (s.d.). The Index of Multiple Deprivation ranges from 0 (most deprived) to 10 (least deprived).

## Table 2 | Change in ADHD-RS total scores over 4 weeks of real TNS versus sham TNS treatment

| Primary outcome (ADHD-RS) | Real TNS (mean, s.d.) | Sham TNS (mean, s.d.) | aMD (95% CI) | Cohen's d (95% CI) | P value |
|---|---|---|---|---|---|
| Baseline | 35.4 (9.7) | 35.2 (9.8) | N/A | N/A | |
| Week 1 | 26.6 (11.8) | 22.9 (11.4) | 3.03 (0.45–5.61) | 0.31 (0.05–0.58) | N/A |
| Week 2 | 25.4 (12.6) | 22.9 (12.3) | 2.30 (−0.25 to 4.84) | 0.24 (−0.03 to 0.50) | N/A |
| Week 3 | 24.1 (11.9) | 22.5 (12.0) | 1.56 (−1.24 to 4.37) | 0.16 (−0.13 to 0.45) | N/A |
| Week 4 | 26.1 (12.3) | 25.0 (12.3) | 0.83 (−2.47 to 4.13) | 0.09 (−0.26 to 0.43) | 0.622 |

CI, confidence interval; Cohen's d, standardized effect size (0.2=small, 0.5=medium, 0.8=large); N/A, not applicable. P values were calculated using two-sided z-tests from the linear mixed models as outlined in methods.

## Secondary outcomes

No significant between-group difference was observed for the ADHD-RS total score at 6-month follow-up (aMD = −0.29; 95% confidence interval: −3.17 to 2.59; $P = 0.845$; Cohen's $d = −0.03$). No significant group differences were observed for most of the other secondary outcomes at week 4 and at 6-month follow-up (Table 3). An exception was the Mind Excessively Wandering Scale (MEWS) total score at week 4, which showed a statistically significant group difference (aMD = −2.17; 95% confidence interval: −4.33 to −0.01; $P = 0.049$; Cohen's $d = −0.27$) in favor of the real TNS group compared to the sham TNS group. Teacher ratings (Conners Teacher Rating Scale short form T-S and ADHD-RS-T) were not analyzed due to high degree of missing data (80%). Similarly, Columbia-Suicide Severity Rating Scale (C-SSRS) scores were not analyzed due to the lack of variation in scores. Descriptive statistics for

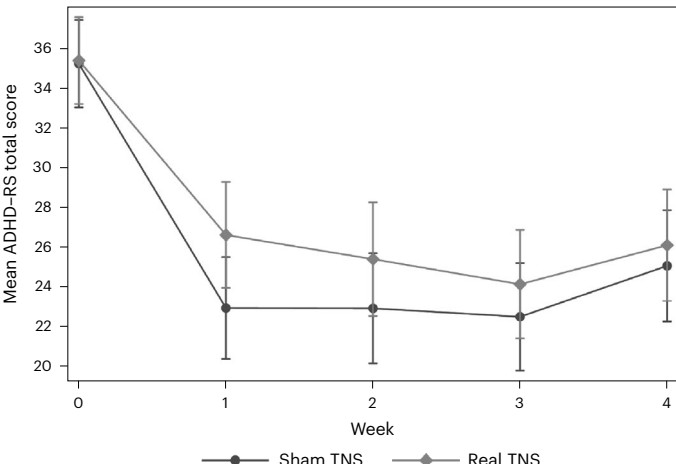

**Fig. 2 | ADHD-RS plot.** Temporal plot of the primary outcome, the investigator-scored, parent-rated ADHD-RS, showing raw means by trial group over time (*n* = 150; real TNS, *n* = 75; sham TNS, *n* = 75) with error bars representing 95% confidence intervals. The ADHD-RS ranges from 0 to 54, with higher scores indicating more severe ADHD symptoms. Scores decreased in both groups across the 4-week treatment period, followed by a slight increase from week 3 to week 4.

teacher ratings and C-SSRS scores at baseline, at week 4 and at 6-month follow-up are presented in Supplementary Table 3.

### Safety
At week 4, the most reported side effects were having trouble sleeping, feeling drowsy/sleepy, headaches and feeling nervous/hyper; however, most of these side effects were rated as mild by participants (Supplementary Table 4) and common in children with ADHD. As illustrated in Table 3, side effect rates and pulse (beats per minute (bpm)) did not differ between treatment groups. The most commonly reported adverse device effects (ADEs) were headaches (real TNS = 21.3%, sham TNS = 17.3%) and difficulties falling asleep or sleep disturbances (real TNS = 20%, sham TNS = 9.3%). Other reported ADEs included physiological symptoms (real TNS = 13.3%, sham TNS = 5.3%), grumpiness or irritability (real TNS = 4%, sham TNS = 5.3%), tearfulness, sadness or depression (real TNS = 1.3%, sham TNS = 6.7%), tiredness, demotivation or joylessness (real TNS = 4%, sham TNS = 1.3%) and frustration (real TNS = 4%, sham TNS = 1.3%) (Extended Data Tables 4–6, Extended Data Fig. 1 and Supplementary Tables 5 and 6). No serious adverse events, serious ADEs or unanticipated serious ADEs were reported, and no participants withdrew from the trial due to adverse events. Six participants discontinued treatment but remained in the trial. Three stopped on their own due to adverse events (nightmares, sleep issues and increased hyperactivity—all in the real group), and three were withdrawn on our clinicians' advice (S.C. and P.S.) for safety reasons (sham: unrelated head injury and emotional sensitivity; real: recurring nosebleeds that stopped shortly after discontinuation).

Most parents (97%) and children (92.5%) reported no or only mild side effects on the acceptability questionnaire administered at week 4. Similarly, most parents (89.8%) and children (82.3%) indicated no or only mild burden on the same questionnaire (Extended Data Table 7).

### Sensitivity analyses
The prespecified complier average causal effect (CACE) analysis showed no significant group difference for the ADHD-RS total score at the week 4 primary endpoint in participants who would comply with the assigned treatment (mean difference = 1.12; 95% confidence interval: −1.38 to 3.61; *P* = 0.381; Cohen's *d* = 0.12), consistent with the ITT analysis (Extended Data Table 8).

The prespecified subgroup analysis of participants who were off medication/medication-naive at baseline found no significant

group difference for the ADHD-RS total score at week 4 (mean difference = 0.68; 95% confidence interval: −3.59 to 4.95; *P* = 0.755; Cohen's *d* = 0.07; Extended Data Table 9).

### Post hoc analyses
For comparability of findings with the previous pilot RCT[20], a post hoc analysis was conducted including only participants in the same age group (8–12 years). No significant group difference was observed for ADHD-RS total score at the week 4 primary endpoint (aMD = 0.55; 95% confidence interval: −3.73 to 4.83; *P* = 0.80) for this subgroup (Supplementary Table 7).

Given our observation that younger children had difficulties in understanding the MEWS items, we conducted a post hoc subgroup analysis in older adolescents (14–18 years), where we were more confident in their comprehension of the scale statements, to test whether we would still observe the effect. No significant group difference was observed for MEWS total score at week 4 (aMD = 0.28; 95% confidence interval: −3.65 to 4.21; *P* = 0.89; Supplementary Table 8).

An additional post hoc subgroup analysis (by editorial request, in accordance with Sex and Gender Equity in Research (SAGER) guidelines (https://ease.org.uk/communities/gender-policy-committee/the-sager-guidelines/)) was conducted disaggregated by sex at birth (Supplementary Tables 9 and 10).

### Discussion
This multicenter, double-blind, randomized, parallel-group, confirmatory phase 2b study tested the efficacy of 4-weeks' nightly use of real versus sham TNS on ADHD clinical symptoms and related problems. We found no differential effects of real versus sham TNS on the primary outcome, the parent-rated ADHD-RS score or any secondary outcomes except for ratings on the MEWS, evaluating mind-wandering, which showed improvement in the real versus sham TNS group. Adherence (93.3%) and reported compliance (93%) were extremely high, likely reflecting the current preference of parents and users for non-pharmacological treatments[12]. Safety was high, with no serious adverse events, and side effects were similar across groups. Acceptability was also high. Blinding was successful.

The findings of this large, double-blind, multicenter RCT do not provide support for TNS as an effective treatment for ADHD. If anything, the sham group had numerically reduced ADHD symptoms on the ADHD-RS at week 1 with an effect size of 0.3. This conflicts with the positive findings from the previous pilot RCT of TNS in children with ADHD that showed an improvement in ADHD symptoms with medium effect size for real versus sham TNS with almost the same protocol with respect to trial duration (4 weeks) and dose/nightly settings[20]. However, a key distinction of our study is the improved design of the sham condition, which is likely responsible for the apparent successful blinding after 4 weeks of treatment. Although the RCT of McGough et al.[20] applied no stimulation at all in the sham TNS condition, in our RCT, the sham TNS group received 30-second stimulation at a lower frequency, followed by 3.570 seconds without stimulation, for every hour of stimulation. This likely improved blinding over the previous RCT. Notably, in the trial of McGough et al.[20], blinding was successful after 1 week. However, participants/parents were not asked about their blinding at the end of the 4-week trial, when unblinding was more likely. Evidence shows that the placebo effect is greater in trials involving technology, such as neurofeedback[29] and neurostimulation, as well as in studies with younger age groups, larger sample sizes, multisite designs and higher baseline symptom severity (ADHD severity was an inclusion criterion in this trial)[30]. This is furthermore enhanced by a nocebo effect in those who realize that they are in the sham condition[31]. In our trial, both groups improved in ADHD symptoms by 26% (real) and 29% (sham). Given that a substantial number of participants in both groups thought that they were in the active condition, the observed effects may reflect a neurotechnology-induced placebo effect or 'neuro-enchantment' or

**Table 3 | Values and statistical comparisons of secondary outcome measures at baseline, 4 weeks and 6 months**

| Secondary outcomes | Mean (s.d.) [n] | | | | | | Effect estimates | | | |
| --- | --- | --- | --- | --- | --- | --- | --- | --- | --- | --- |
| | Baseline | | Week 4 | | Month 6 | | Week 4 | | Month 6 | |
| | Real TNS (n=75) | Sham TNS (n=75) | Real TNS (n=73) | Sham TNS (n=74) | Real TNS (n=72) | Sham TNS (n=73) | aMD (95% CI) | p-value | aMD (95% CI) | p-value |
| SDQ Hyperactivity/ impulsivity/ inattention score (child rated) | 7.5 (2.0) [75] | 7.5 (2.1) [75] | 6.8 (2.3) [73] | 7.0 (2.1) [74] | 6.6 (2.2) [71] | 6.7 (2.1) [73] | -0.30 (-0.88, 0.28) | 0.308 | -0.24 (-0.83, 0.36) | 0.433 |
| ARI-P total score (parent rated) | 5.5 (3.3) [75] | 5.0 (3.2) [75] | 3.8 (3.2) [73] | 3.9 (3.1) [74] | 4.6 (3.1) [72] | 4.4 (3.2) [73] | -0.36 (-1.15, 0.43) | 0.374 | -0.01 (-0.83, 0.80) | 0.974 |
| ARI-S total score (child rated) | 4.2 (3.4) [75] | 4.0 (3.3) [75] | 2.9 (3.2) [73] | 3.4 (3.2) [74] | 3.3 (2.8) [71] | 3.2 (3.0) [73] | -0.63 (-1.27, 0.01) | 0.052 | -0.11 (-0.83, 0.61) | 0.766 |
| MEWS total score (child rated) | 16.7 (8.1) [75] | 17.3 (8.2) [75] | 13.4 (8.9) [73] | 15.9 (9.8) [74] | 15.0 (9.6) [71] | 15.9 (8.9) [73] | -2.17 (-4.33, -0.01) | 0.049* | -0.73 (-3.15, 1.68) | 0.553 |
| RCADS-25 total score (child rated) | 41.6 (9.0) [74] | 42.7 (10.4) [75] | 36.9 (7.1) [73] | 39.1 (9.1) [74] | 38.6 (8.5) [71] | 40.7 (10.0) [73] | -1.56 (-3.54, 0.41) | 0.121 | -1.45 (-4.04, 1.15) | 0.274 |
| RCADS-25 total score (parent rated) | 58.5 (12.6) [73] | 56.9 (13.7) [75] | 50.3 (10.1) [73] | 50.7 (11.7) [74] | 55.4 (14.2) [72] | 53.3 (11.9) [73] | -1.07 (-3.87, 1.73) | 0.453 | 1.41 (-1.95, 4.77) | 0.410 |
| Mackworth Vigilance Task (% of omission errors) | 45.8 (23.8) [75] | 41.4 (21.7) [74] | 36.3 (21.1) [70] | 30.1 (21.8) [74] | 28.3 (17.0) [64] | 25.8 (19.9) [72] | 3.62 (-0.73, 7.98) | 0.103 | -0.15 (-4.90, 4.60) | 0.950 |
| Mackworth Vigilance Task (% of commission errors) | 6.8 (8.6) [73] | 5.9 (6.6) [75] | 4.7 (5.5) [70] | 6.8 (12.8) [74] | 4.2 (7.0) [64] | 5.7 (10.4) [72] | 0.95[1] (0.80, 1.13) | 0.573 | 0.90 (0.73, 1.10) | 0.283 |
| SDSC total score (parent rated) | 49.2 (12.1) [75] | 44.0 (9.7) [74] | 43.2 (10.0) [73] | 39.6 (8.9) [74] | 46.1 (12.1) [72] | 42.7 (9.7) [72] | 1.00 (-1.42, 3.42) | 0.417 | 0.51 (-2.46, 3.47) | 0.738 |
| Objective hyperactivity composite score[1] | −0.1 (1.7) [73] | 0.1 (1.7) [74] | −0.2 (1.8) [71] | 0.2 (1.7) [73] | N/A | N/A | -0.25 (-0.74, 0.24) | 0.319 | N/A | N/A |
| Average pupil diameter at rest | 9.5 (1.5) [75] | 9.5 (1.6) [75] | 9.0 (1.3) [72] | 9.3 (1.6) [74] | 9.3 (1.4) [65] | 9.7 (1.6) [69] | -0.24 (-0.56, 0.07) | 0.133 | -0.29 (-0.64, 0.06) | 0.0100 |
| Average pupil diameter at task | 9.9 (1.6) [75] | 10.1 (1.5) [75] | 9.5 (1.3) [72] | 9.8 (1.6) [74] | 9.7 (1.5) [65] | 10.0 (1.4) [70] | -0.17 (-0.52, 0.18) | 0.332 | -0.16 (-0.54, 0.21) | 0.314 |
| Side effects score (child rated) | 12.5 (10.1) [75] | 12.3 (9.3) [75] | 10.7 (9.5) [75] | 11.9 (10.5) [75] | 7.5 (7.6) [70] | 8.5 (7.1) [73] | -1.11 (-3.76, 1.53) | 0.410 | -1.05 (-3.09, 0.99) | 0.314 |
| Side effects score (parent rated) | 10.8 (7.9) [75] | 8.9 (6.5) [75] | 8.7 (6.0) [75] | 9.0 (6.8) [75] | 7.6 (6.2) [71] | 7.2 (6.4) [73] | -1.09 (-2.79, 0.61) | 0.210 | -0.26 (-2.03, 1.52) | 0.777 |
| Weight (kg) | 46.7 (13.8) [75] | 47.4 (14.4) [75] | 47.2 (14.1) [72] | 48.1 (14.8) [74] | 49.3 (14.4) [65] | 49.7 (15.0) [71] | -0.39 (-0.83, 0.05) | 0.080 | 0.13 (-0.68, 0.94) | 0.754 |
| Pulse (bpm) | 78.4 (13.7) [75] | 78.5 (14.2) [74] | 78.0 (12.5) [72] | 79.4 (13.7) [74] | 78.5 (13.0) [65] | 78.1 (14.1) [71] | -1.32 (-4.97, 2.32) | 0.477 | 0.79 (-2.96, 4.54) | 0.679 |

ARI-P, Affective Reactivity Index-Parent Report; ARI-S, Affective Reactivity Index-Self Report; kg, kilograms; N/A, not applicable. *P* values were calculated using two-sided *z*-tests from the linear mixed models as outlined in methods. No *P* value adjustment was made for multiple outcomes. [1]Reported beta estimate is back transformed after log transformation of this outcome due to skewness of residuals. The back-transformed estimate given here is a geometric mean ratio for this outcome instead of a mean difference. *Significant at *P* < 0.05.

'neuro-suggestion'[32]. In fact, the sham group improved by 10 points on the ADHD-RS, which is equivalent to a large Cohen's *d* of 0.9, which is more than double the pooled medium effect size of 0.4 for parent ratings of the ADHD-RS reported in a meta-analysis of 27 RCTs of medication and placebo effects in ADHD[33]. Our findings, hence, extend previous evidence in the literature[29,30] that the placebo response related to a neurotechnology such as TNS is larger than the typical placebo response in medication trials. Alternative explanations are a regression to the mean, potential baseline severity symptom inflation, as parents were aware of severity criteria for trial entry, or non-specific beneficial effects of staff interaction[30]. It could also potentially be argued that sham conditions sharing features with the intervention may dilute its effects and, hence, compromise its validity. However, the sham stimulation had lower frequency and pulse width, and it is unlikely that 30 seconds of stimulation every hour with such low frequency and pulse width would have led to an improvement in symptoms.

Another difference with respect to the previous pilot RCT[20] is that we included long-term medicated children (39.3%) and a larger age range of children and adolescents of 8–18 years, whereas the previous study was restricted to non-medicated children (8–12 years). Medication could potentially mask effects or interact with TNS. However, our subgroup analysis in non-medicated children and adolescents also showed no effect nor did a post hoc analysis in the same age range as the one used in the previous pilot study[20] (Extended Data Table 9 and Supplementary Table 7).

The only positive finding of real versus sham TNS was an improvement in the MEWS mind-wandering scale after 4 weeks. Mind-wandering has been found to be a key behavioral impairment in people with ADHD, which is thought to interfere with cognitive/attention performance[27]. This is further underpinned by consistent evidence at the brain level for increased activation in people with ADHD of the default mode network, which mediates mind-wandering, during cognitive and attention task performance and during rest[23,24,34,35] and by evidence for a poor anti-correlation between the default mode network and attention networks in people with ADHD relative to healthy controls[35].

Given that increased mind-wandering is a core feature of ADHD[27,36,37], this may represent a clinically meaningful benefit of the treatment. However, this needs to be considered in the context of

negative findings in all other 16 measures, and a possible type I error due to multiple testing. Also, the younger children in the trial had difficulty understanding the MEWS items, which, although validated in children[38], were originally designed for adults with ADHD[39]. A post hoc analysis of older adolescents aged 14–18 years whom we are more confident understood the MEWS items, however, showed no effect (Supplementary Table 8).

The RCT also showed no effect on a key measure of vigilance/ sustained attention that is typically impaired in children with ADHD. Although an open-label pilot study of TNS reported a significant reduction in flanker task incongruent reaction times to incongruent trials in the flanker task (that is, reaction times to incongruent trials are typically slower than those to congruent trials, which is an indicator of interference inhibition) after 8 weeks of treatment[26], this finding was not replicated in the subsequent double-blind pilot RCT[20]. In that trial[20], only participants classified as TNS responders showed reductions in behavioral measures of working memory, which predicted treatment response and correlated with symptom improvement. However, performance on computerized cognitive tasks, including working memory and Stroop tests, did not predict treatment response[40].

We also found no effect on objective wrist-held measures of hyperactivity nor on pupil diameter, a key physiological measure of arousal and autonomic nervous system (ANS) activity. The lack of treatment-induced pupil dilation suggests that TNS may not significantly influence the ANS, thus challenging its proposed bottom-up mechanisms of action through the locus coeruleus and brainstem[14,16].

In line with our previous meta-analysis of TNS across neurological and psychiatric conditions[19], and the earlier pilot study of TNS in children with ADHD[20], safety was excellent, with no group differences in side effects and no serious adverse events. Acceptability was also excellent, with most participants reporting mild or no burden. TNS is, hence, very safe and tolerable but, unfortunately, not effective for youth with ADHD. The study population was very representative of the general UK population in terms of race/ethnicity, with 79.3% identifying as White (81% in the Census 2021; https://www.ons.gov.uk/ peoplepopulationandcommunity/culturalidentity/ethnicity/bulletins/ ethnicgroupenglandandwales/census2021), and 20.7% from other ethnic groups (18.7% in the Census 2021).

Although this RCT study was rigorously conducted, with an improved and more rigorous control condition over previous trials[20,26], it had some limitations.

Limitations include a high rate of missing data on teacher ratings (80%) due to low teacher participation. As we did not have the power to analyze teacher ratings, it was not possible to investigate potential treatment-related changes in participants' inattentive and/or impulsive/hyperactive behaviors within school settings. Parent ratings are subject to various biases, including those related to parental stress and demographic factors[41,42].

Also, although adherence was very high (93.3%), it was self-reported and may have been overestimated due to social desirability bias[43]. Adherence relied on participant-completed nightly sleep diaries to track device use. Unfortunately, these could not be corroborated by objective device-logged usage data as they were found not to be reliable. Future studies should incorporate reliable and accurate objective device usage monitoring to improve the accuracy of adherence assessment and ensure treatment fidelity.

The inclusion of medication could have been a confound, but, as discussed above, the effects remained the same in non-medicated participants.

In summary, this rigorously controlled multicenter RCT found that, despite high compliance and adherence (of over 93%), 4 weeks of nightly TNS did not improve core symptoms or related clinical and cognitive features in children and adolescents with ADHD. These negative findings on TNS extend largely negative findings using other neurostimulation techniques in children and adults with ADHD, including

transcranial magnetic and direct current stimulation[13,44–48]. This large, multicenter RCT contrasts with the positive symptom improvements reported in the pilot trial that informed FDA clearance for TNS[20], highlighting the critical importance of robust sham control conditions and expectation management to minimize placebo effects in neurostimulation research. In conclusion, although TNS is a safe intervention, it does not demonstrate clinical efficacy for pediatric ADHD.

## Online content

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

[1]Department of Child & Adolescent Psychiatry, Institute of Psychiatry, Psychology & Neurosciences King's College London, London, UK. [2]Florence Nightingale Faculty of Nursing, Midwifery & Palliative Care, Division of Care in Long Term Conditions, King's College London, London, UK. [3]Developmental EPI (Evidence synthesis, Prediction, Implementation) Lab, Centre for Innovation in Mental Health, School of Psychology, Faculty of Environmental and Life Sciences, University of Southampton, Southampton, UK. [4]Department of Psychology, Manisa Celal Bayar University, Manisa, Turkey. [5]Department of Biostatistics and Health Informatics, Institute of Psychiatry, Psychology & Neuroscience, King's College London, London, UK. [6]King's Clinical Trials Unit, Institute of Psychiatry, Psychology & Neuroscience, King's College London, London, UK. [7]Attention Deficit Disorder Information and Support Services (ADDISS), Edgware, UK. [8]Mental Health Dresden-Leipzig, Technical University Dresden, Dresden, Germany. [9]The Giaroli Centre, London, UK. [10]The Assessment Team, Hertfordshire, UK. [11]The Autism Research Centre, The University of Cambridge, Cambridge, UK. [12]Centre for Interventional Paediatric Psychopharmacology and Rare Diseases (CIPPRD), National and Specialist CAMHS, South London and Maudsley NHS Foundation Trust, London, UK. [13]Centre for Interventional Paediatric Psychopharmacology (CIPP) Rett Centre, Institute of Psychiatry, Psychology and Neuroscience, King's College London, London, UK. [14]Department for Neuroimaging, Institute of Psychiatry, Psychology & Neuroscience, King's College London, London, UK. [15]Clinical and Experimental Sciences (CNS and Psychiatry), Faculty of Medicine, University of Southampton, Southampton, UK. [16]Hampshire and Isle of Wight Healthcare NHS Foundation Trust, Southampton, UK. [17]Hassenfeld Children's Hospital at NYU Langone, New York University Child Study Center, New York, NY, USA. [18]DiMePRe-J-Department of Precision and Regenerative Medicine-Jonic Area, University of Bari 'Aldo Moro', Bari, Italy. [19]These authors contributed equally: Aldo Alberto Conti, Natali Bozhilova, Irem Ece Eraydin. ✉e-mail: katya.rubia@kcl.ac.uk

## Methods

### Trial design

This UK multicenter (King's College London and University of Southampton), phase 2b, double-blind, parallel group, sham-controlled confirmatory RCT was preregistered (trial registration: ISRCTN82129325; date of registration: 8 February 2021). Participants were randomized to either active TNS or sham TNS (1:1). For protocol details, see ref. 28.

The trial was approved by the West Midlands–Solihull NHS Research Ethics Committee (REC; Ref21:/WN/0169; IRAS: 299703) and the Medicines and Healthcare products Regulatory Agency (MHRA; Ref: CI/2022/0003/GB). It was conducted in accordance with the 1975 Declaration of Helsinki and is reported following CONSORT guidelines[49]. Independent oversight of the trial was provided by a Data Monitoring and Ethics Committee and a Trial Steering Committee. The lead of Patients and Public Involvement (PPI) co-author (A.B.), an expert user group at King's College London and an ethnically diverse group of patients and their parents co-designed the study to make it as equitable, diverse and inclusive as possible and gave advice and assisted with recruitment and dissemination throughout the study.

### Randomization and blinding

Randomization was done by minimization by sex (male/female), medication status (on medication, off medication/naive), site (London, Southampton) and age (8–13.5 years, 13.6–19 years) using a validated, online, web-based system from King's Clinical Trials unit[28]. Participants, parents/carers, postdoctoral research associates, principal investigator, co-investigators and analysts were blinded to treatment group except for the trial manager (L.J.) and trial manager assistants (S.E.M. and J.H.), who trained participants/parents on the device use but did not conduct research assessments and were prohibited from sharing the information with other team members. Analysts were blinded until after database lock. Blinding was assessed by a questionnaire administered to participants, parents/carers and researchers after 1 week and 4 weeks of TNS treatment.

### Participants

One hundred and fifty children and adolescents (8–18 years at consent stage) with ADHD were recruited from public and private clinics in (Greater) London, Southampton and Portsmouth; from nationwide parent and ADHD support groups; general practitioners; from the National Health System Consent for Contact research directory; and from social media. Inclusion criteria were as follows: a clinical and/or research DSM-5 ADHD diagnosis (semi-structured interview: Kiddie-Schedule for Affective Disorders and Schizophrenia (K-SADS))[50]; a score of ≥24 on the investigator-scored parent-rated ADHD-RS; IQ above 70 (Wechsler Abbreviated Scale of Intelligence (WASI-II))[51]; being able to speak sufficient English (parents and children); and being medication-naive, willing to come off their stimulant medication for 1 week before participation or willing to be on stable stimulant medication for the 4-week RCT duration. Exclusion criteria were as follows: comorbidity with any major psychiatric disorder as assessed on the K-SADS (except for conduct/oppositional defiant disorder, mild anxiety and mild depression, which scored below threshold on the K-SADS); enuresis and encopresis; alcohol and substance abuse; neurological abnormalities; traumatic brain injury (TBI); any other non-pharmacological treatments; dermatitis; and TNS contraindications such as implanted cardiac or neurostimulation systems, head-implanted metallic or electronic devices and body-worn devices. Participants were also excluded if they were medicated with non-stimulants such as atomoxetine, guanfacine or clonidine. Non-stimulant medications have shown to enhance noradrenaline in frontal and cortical regions via selectively blocking noradrenaline transporters (atomoxetine) or by stimulating postsynaptic a2-adrenergic receptors (guanfacine and clonidine)[52]. Given that a key mechanism of action of TNS is thought to be the stimulation of the locus coeruleus, which releases noradrenaline into the brain[15], we excluded these medications due to their similar underlying mechanisms of action to TNS[14,16] and potential interaction effects.

Children/adolescents and their parents/carers provided both digital and written informed consents/assents and were reimbursed for travel costs and received up to £350 (£450 if they were enrolled in the fMRI substudy). For details, see the protocol (ref. 28).

### Procedures

Participants were screened for eligibility via two online appointments and one in-person appointment at King's College London Institute of Psychiatry, Psychology and Neuroscience or the University of Southampton Centre for Innovation and Mental Health. Online screening included study information sheets and device explanations, digital consents and parent/carer K-SADS interviews. In-person screening included IQ testing (WASI-II), child/adolescent K-SADS interviews, mock fMRI, fMRI task training and written informed consents/assents.

During the 2–3-hour baseline assessment, participants and parents/carers completed measures of ADHD symptoms, depression and anxiety, sleep, mind-wandering, emotional dysregulation and suicidality. Children/adolescents performed neurocognitive tasks (30–40 minutes)[53], underwent pupillometry during one of the tasks and wore an Empatica E4 wristband (Empatica Srl) to assess objective hyperactivity and autonomic functions. Vital signs and anthropometrics (height and weight) were recorded, and those participants who enrolled in the fMRI substudy underwent a 1-hour scan. Teachers were contacted prior to the assessment to provide ADHD ratings. Participants were randomized (1:1) to active or sham TNS at the end of the baseline assessment, and both parents/carers and participants were instructed on device use and daily sleep diaries for the 4-week treatment.

Weekly online assessments (20–30 minutes) included ADHD ratings from parents/carers, side effect and adverse event reporting and a blinding questionnaire completed during the week 1 assessment.

At week 4 (2–3 hours, in-person), participants returned the TNS device and repeated baseline tasks. Weight, hyperactivity and vital signs were reassessed, and acceptability and blinding questionnaires were completed. Participants who underwent an fMRI scan at baseline also underwent an fMRI scan during this assessment. Teachers were asked to provide ADHD ratings.

The 6-month follow-up (1–2 hours, in-person) replicated previous baseline and week 4 assessments, except for fMRI and Empatica E4 measurements. Concomitant medications were recorded throughout the trial.

Study data were entered and managed using the MACRO Electronic Data Capture system (version 4.15.0.116).

### Intervention

Real and sham TNS was performed with the Monarch TNS System (NeuroSigma, Inc.). Participants needed to use the stimulator for approximately 8 hours during sleep. Each night, participants or their parents applied the disposable self-adhesive patch electrodes, connected to the stimulator, across their child's forehead to provide bilateral stimulation of V1 trigeminal nerve branches. The real TNS used 120-Hz repetition frequency with a 250-μs pulse width and a duty cycle of 30 seconds on/30 seconds off (total 240 minutes in 8 hours). Stimulator settings were established at baseline (and adjusted each night) by titration in 0.2-mA increments ranging from 0 to a safe maximum of 10 mA to identify a stimulation level that was perceptible but below the participants' subjective level of pain/discomfort. The sham Monarch TNS system was identical in current, appearance and user interface, but the electrical stimulation flowed for 30 seconds every hour at a lower frequency (2 Hz) and 50-μs pulse width and was then routed through the internal resistor instead of the electrical patch, thus still draining battery to maintain blinding (total 4 minutes in 8 hours). The 30 seconds of real stimulation every hour in the sham condition was added to further enhance blinding[28], which was successful in the

previous trial without any stimulation in the sham condition[20]. The scalp adjusts very quickly to the stimulation, and the switch-off is not noticeable. To further protect blinding, participants were counseled that stimulation may not be perceptible and that most people would not feel the stimulation after some time because of scalp adaptation. Technical support was provided by the trial manager (L.J.). For details, see the protocol (ref. 28).

## Safety

Safety was assessed through a weekly side effect questionnaire adapted for TNS[20], a weekly open-ended adverse event form completed by participants and their parents/carers and vital signs (blood pressure and pulse) measured at baseline, at week 4 and at 6-month follow-up.

## Outcome measures

The primary outcome measure was the investigator-scored, parent-rated ADHD-RS total score[21], collected at eligibility, baseline and weekly throughout the 4-week trial. Secondary outcome measures were collected at baseline, at week 4 and at 6-month follow-up and included the following rating scales: teacher-rated ADHD-RS (school version)[54], Conners Teacher Rating Scale short form T-S[55], child-reported Strength and Difficulties Questionnaire (SDQ)[56], parent and child-reported Affective Reactivity Index (ARI)[57], parent and child-reported Child and Adolescent Anxiety and Depression scale (RCADS-25)[58], child-reported C-SSRS[59], child-reported MEWS[39], parent-reported Sleep Disturbance Scale for Children (SDSC)[60] and the investigator-scored, parent-rated ADHD-RS[21] at 6-month follow-up. Vigilance (omission and commission errors) was assessed using the Mackworth Clock Task[61]. Pupillometry data were recorded with the Tobii Pro Nano screen-based eye-tracking device (Tobii AB, Tobii Pro Lab version 1.207) during a 1-minute resting condition and a cognitive task. Objective hyperactivity, defined as the composite score of both the intensity ($g$) and frequency ($g$) of movement, was assessed at baseline and week 4 using a three-axis accelerometer embedded in the Empatica E4 wristband device (Empatica Srl, version 2.0.3 (5119)). Other measures included an acceptability questionnaire filled out by participants and their parents/carers at the end of the treatment.

Details regarding other secondary outcome measures of executive functions, physiological and fMRI measures are described in the study protocol[28] and will be published elsewhere.

## Sample size justification

The estimated sample size of 128 participants (64:64) was calculated using a baseline to posttreatment correlation of 0.5, 90% power, 5% type I error and an anticipated effect size of 0.5 for a reduction in ADHD symptoms at 4 weeks[20]. The number of participants was inflated to 150 (75:75) to account for a loss to follow-up rate of 15%[28].

## Statistical analysis

Analyses were performed in Stata 18 (StataCorp LLC, version 18.0) following a prespecified statistical analysis plan, which can be found as supplementary material to the published protocol[28].

For the primary analysis, a longitudinal linear mixed model was used, fitting 4-week ADHD symptom scores as a continuous outcome, with continuous time as a covariate using actual observed time of assessments and an interaction between time and trial group to estimate treatment effects at week 1, 2, 3 and 4 using post-estimation. A random intercept was included as well as a random slope over time for each participant and assuming an independent covariance matrix for these random effects. We additionally adjusted for fixed effects of baseline ADHD-RS score, site (London, Southampton), age category (8–13.5 years, 13.6–19 years), sex at birth (male, female) and medication status (on stable medication, off medication/naive). An aMD was calculated between the treatment groups with associated 95% confidence intervals and $P$ value (for week 4 only). A separate model was used to

investigate treatment differences at the 6-month timepoint by including time as a categorical variable, as treatment differences at follow-up were not expected to follow the same linear time trend. For the analysis of secondary outcomes, we used mixed models for repeated measures (MMRM) with time included as a categorical variable and the same covariates as for the primary analysis. An ITT approach was used for both primary and secondary analyses. No adjustment for multiple timepoints was performed as we prespecified the primary outcome at week 4 (ref. 62). Statistical significance for all analyses was $P < 0.05$. Cohen's $d$ was calculated using the pooled baseline standard deviation of each measure.

A separate analysis of the primary outcome was carried out to estimate the treatment effect in those participants who adhered to the intervention, using a CACE analysis. Further details on the statistical methods can be found in the supplement and the protocol paper and its supplement[28].

## Reporting summary

Further information on research design is available in the Nature Portfolio Reporting Summary linked to this article.

## Data availability

All data supporting the findings of this study have been deposited in the figshare repository and are publicly available at https://doi.org/10.6084/m9.figshare.29414744.v1. Source data underlying the figures and tables presented in this paper are included in the figshare repository[63]. No custom code was generated or used in this study.

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

## Acknowledgements

This project was funded by the Efficacy and Mechanism Evaluation (EME) Programme (NIHR130077) to K.R., a Medical Research Council (MRC) and National Institute for Health and Care Research (NIHR) partnership (NIHR130077). K.R., B.C. and M.A.M. are also supported by the NIHR Biomedical Research Centre (BRC) at South London and Maudsley National Health Service (NHS) Foundation Trust and King's College London (NIHR BRC Maudsley). K.R. is also supported by an NIHR grant (NIHR203684), the MRC (APP32868), the Medical Research Foundation (MRF-176-0002-RG-FLOH-C0929) and the Rosetrees Foundation (3442198). S.C. is an NIHR Research Professor (NIHR303122) funded by the NIHR for this research project. S.C. is also supported by the NIHR (NIHR203684, NIHR203035, NIHR128472 and RP-PG-0618-20003) and by a grant (101095568-HORIZONHLTH-2022-DISEASE-07-03) from the European Research Executive Agency. P.S. has received funding from the NIHR (NIHR203684 and NIHR131175), the British Heart Foundation (SP/F/21/150013) and GUTS UK (DGN2019_13). B.C. has received funding from the NIHR (17/32/04, NIHR130674, NIHR130730, NIHR127921, NIHR158490, NIHR165421, NIHR207448, NIHR154546 and NIHR127951), the Department of Education, the Nuffield Foundation (EDO/FR-000022056) and Alzheimer's Research UK (ARUK-GCTF2018B-001).

A.B. is further supported by grants from the NIHR (NIHR203684 and NIHR207448), the MRC (MR/Y003209/1 and APP32868) and European Union Horizon 2020 (965381). The design, management, analysis and reporting of the study are independent of the funders and the device manufacturers, and the views expressed in this publication are those of the authors and not necessarily those of the MRC, the NIHR or the Department of Health and Social Care or any of the other funding bodies.

This study is supported by the NIHR Applied Research Collaboration South London at King's College Hospital NHS Foundation Trust.

This study was supported by the UK Clinical Research Collaboration-registered King's Clinical Trials Unit (KCTU) at King's Health Partners, which is partly funded by the NIHR BRC Maudsley for Mental Health, hosted by South London and Maudsley NHS Foundation Trust and King's College London, and the NIHR Evaluation, Trials and Studies Coordinating Centre. This study was supported by the KCTU with a bespoke InferMed randomization and MACRO electronic data capture system, alongside the KCTU quality management system. A particular thanks to M. Martin who helped with data analysis.

Thanks to all the participants and their parents/carers for their time and support for the project. Recruitments were made through the CAMHS clinics within the following NHS trusts: South London and Maudsley NHS Foundation Trust, Hampshire and Isle of Wight Healthcare (previously known as SOLENT NHS Trust), Central and North-West London NHS Foundation Trust, Oxleas NHS Foundation Trust and South-West London and St. George's Mental Health NHS Trust. In addition, recruitment was made by private clinics from the Giaroli Centre and the Assessment Team. The project received support from the South London Regional Research Delivery Network, in particular K. O'Brien. Many thanks to ADHD Foundation Trust and ADHD Information Services, which advertised the study on their social media, as well as other local ADHD support groups in the South-East of England. Valuable advice and support were provided throughout the trial by the study-dedicated PPI group (M. Reilly, co-ordinator Lambeth ADHD Support Group; F. Mahamed Ali; Z. Hassan Ali; H. Reynolds; and J. Reynolds). In addition, further help was given by S. Zahid and L. Arturi. We are also grateful for the advice and support given throughout the trial by the Data Monitoring and Ethics Committee (D. Daley, A. Sharma and A. Cook) and the Trial Steering Committee (R. Morriss, J. Warner-Rogers, A. James, R. Evans, B. Nolker and M. Nolker).

## Author contributions

Grant funding was obtained by K.R. (principal investigator (PI)) and S.C., B.C., P.S., M.A.M. and A.B. (co-investigators) who conceived and designed the study. S.C. was the PI lead for the Southampton site. A.A.C. and N.B. recruited and conducted assessments of all participants at the London site. I.E.E. recruited and conducted assessments of all participants at the Southampton site and was assisted by R.M. L.J. managed all aspects of the trial across both sites and also administered device training and technical support, which was assisted by S.E.M. and J.H. D.S. and B.C. wrote the statistical analysis plan, which was edited by K.R. and L.J., analyzed the data and wrote the statistical report. S.C. and P.S. were responsible for all clinical aspects of the study. K.R., S.C., B.C., P.S., M.A.M., A.B., L.J., A.A.C., N.B. and I.E.E. were responsible for the interpretation of the data. G.G. and H.L. assisted with recruitment. F.F. analyzed the Empatica E4 physiological data, under the supervision of P.S. A.A.C., N.B., I.E.E. and K.R. wrote the first draft of the manuscript. All authors contributed to, revised and approved the final version of the manuscript.

## Competing interests

S.C. has received reimbursement for travel and accommodation expenses from the Association for Child and Adolescent Central Health (ACAMH) in relation to lectures delivered for ACAMH, the Canadian AADHD Alliance Resource, the British Association of Psychopharmacology, Healthcare Convention and CCM Group team for educational activity on ADHD and has received honoraria from Medice. M.A.M. has received research funding from Takeda Pharmaceuticals, Johnson and Johnson, Lundbeck, Boehringer Ingelheim and Nxera. He also acted as a consultant for Neurocrine, Lundbeck, Boehringer Ingelheim and Nxera. G.G. has received speaker honoraria from Takeda Pharmaceuticals. P.S. reports research funding from the British Heart Foundation, Reverse Rett, Newron Pharmaceuticals, HealthTracker and Anavex Life Sciences Corporation; consulting fees from Anavex Life Sciences Corporation; honoraria and reimbursement for travel and accommodation expenses from the Egyptian Psychiatry Association, Acadia Pharmaceuticals, Inc. and Neurogene, Inc.; and stocks from HealthTracker, Ltd. F.F. is a shareholder and the Chief Technical Officer of HealthTracker, Ltd. B.C. has received funding from LifeARC and Mundipharma. K.R., D.S., A.A.C., N.B., I.E.E., R.M., A.B., S.E.M., J.H., H.L. and L.J. have no financial interests to declare.

## Additional information

**Extended data** is available for this paper at https://doi.org/10.1038/s41591-025-04075-x.

**Correspondence and requests for materials** should be addressed to Katya Rubia.

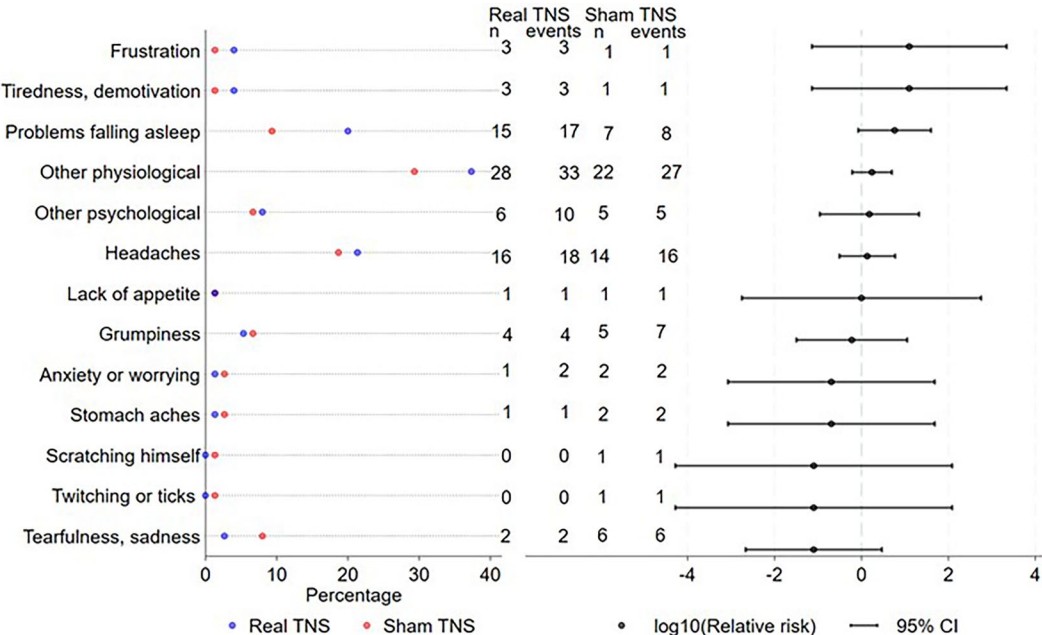

**Extended Data Fig. 1 | Dot plot of adverse events by trial arm with log relative risks. Legend:** Full labels for adverse event categories where applicable: **Tiredness, demotivation**, joylessness; **Problems falling asleep** or sleep problems; **Other physiological** event, **Other psychological** event; **Grumpiness** and irritability; **Scratching himself**/herself, biting nails or lips more; **Twitching or ticks** (eye blinking, head tics); **Tearfulness, sadness** or depression. Left sided graph gives percentage of participants who experienced that adverse event type in each arm. "n" gives number of participants who experienced event in that trial arm, "events" gives total number of that type of events in that trial arm. Right sided graph gives log relative risks (Real TNS vs Sham TNS) of each event type with bars indicating 95% CIs (Confidence intervals). Dot plot uses package by Rachel Phillips & Suzie Cro, 2020. "AEDOT: Stata module to produce dot plot for adverse event data," Statistical Software Components S458735, Boston College Department of Economics, revised 17 Nov 2021.

**Extended Data Table 1 | Type of stimulant medications, mean daily dose and dose ranges for stimulant medications recorded at baseline**

| Medication at baseline | Methylphenidate | | Amphetamine | |
|---|---|---|---|---|
| | N (%) | Mean daily dose (SD) (mg) [range] | N (%) | Mean daily dose (SD) (mg) [range] |
| Real TNS (N=75) | 26 (34.7%) | 22.8 (15.9) [5-90] | 3 (4.0%) | 35.0 (12.9) [20-50] |
| Sham TNS (N=75) | 22 (29.3%) | 30.2 (17.3) [5-72] | 8 (10.7%) | 33.2 (17.6) [5-60] |

**Extended Data Table 2 | Concomitant medications at baseline, week 4 and at 6 months follow-up**

| Medication Category (n participants, %) | Baseline | | Week 4 | | Month 6 | |
|---|---|---|---|---|---|---|
| | Real TNS | Sham TNS | Real TNS | Sham TNS | Real TNS | Sham TNS |
| Stimulant medication | 29 (38.7) | 30 (40.0) | 29 (38.7) | 30 (40.0) | 31 (41.3) | 35 (46.7) |
| Other psychotropic medication | 9 (12.0) | 10 (13.3) | 9 (12.0) | 11 (14.7) | 9 (12.0) | 16 (21.3) |
| Other medication type | 8 (10.7) | 12 (16.0) | 13 (17.3) | 18 (24.0) | 14 (18.7) | 18 (24.0) |

**Extended Data Table 3 | Parent, child and researcher blinding guesses at week 1 and week 4**

| Blinding guesses | Child | | Parent | | Researcher | |
|---|---|---|---|---|---|---|
| | Real TNS | Sham TNS | Real TNS | Sham TNS | Real TNS | Sham TNS |
| Week 1 - guess, N (%) | | | | | | |
| Real TNS | 34 (45.3) | 34 (45.3) | 30 (40.0) | 16 (21.3) | 8 (10.7) | 9 (12.0) |
| Sham TNS | 9 (12.0) | 13 (17.3) | 12 (16.0) | 17 (22.7) | 8 (10.7) | 12 (16.0) |
| Don't know | 32 (42.7) | 28 (37.3) | 33 (44.0) | 42 (56.0) | 59 (78.7) | 54 (72.0) |
| Week 4 - guess, N (%) | | | | | | |
| Real TNS | 28 (37.8) | 24 (32.0) | 28 (37.8) | 16 (21.3) | 10 (13.5) | 10 (13.3) |
| Sham TNS | 21 (28.4) | 22 (29.3) | 24 (32.4) | 29 (38.7) | 21 (28.4) | 27 (36.0) |
| Don't know | 24 (32.4) | 28 (37.3) | 21 (28.4) | 29 (38.7) | 42 (56.8) | 37 (49.3) |

**Extended Data Table 4 | Adverse event type**

| AE Category | Real TNS | | Sham TNS | |
|---|---|---|---|---|
| | N events | N with event (%) | N events | N with event (%) |
| Non-device related adverse events | 30 | 26 (34.7) | 35 | 22 (29.3) |
| Adverse device effects (ADES) | 67 | 42 (56.0) | 49 | 32 (42.7) |

**Extended Data Table 5 | Overall adverse event category by group from randomisation to week 4**

| AE Category | Real TNS | | | | Sham TNS | | | |
|---|---|---|---|---|---|---|---|---|
| | AES | | ADES | | AES | | ADES | |
| | N events | N with event (%) | N events | N with event (%) | N events | N with event (%) | N events | N with event (%) |
| Anxiety or worrying | 0 | 0 (0.0) | 2 | 1 (1.3) | 1 | 1 (1.3) | 1 | 1 (1.3) |
| Tearfulness or sadness | 1 | 1 (1.3) | 1 | 1 (1.3) | 1 | 1 (1.3) | 5 | 5 (6.7) |
| Tiredness or demotivation | 0 | 0 (0.0) | 3 | 3 (4.0) | 0 | 0 (0.0) | 1 | 1 (1.3) |
| Withdrawal/ less socializing | 0 | 0 (0.0) | 0 | 0 (0.0) | 0 | 0 (0.0) | 0 | 0 (0.0) |
| Grumpiness and irritability | 1 | 1 (1.3) | 3 | 3 (4.0) | 2 | 1 (1.3) | 5 | 4 (5.3) |
| Twitching or ticks | 0 | 0 (0.0) | 0 | 0 (0.0) | 0 | 0 (0.0) | 1 | 1 (1.3) |
| Scratching or biting oneself | 0 | 0 (0.0) | 0 | 0 (0.0) | 0 | 0 (0.0) | 1 | 1 (1.3) |
| Headaches | 0 | 0 (0.0) | 18 | 16 (21.3) | 1 | 1 (1.3) | 15 | 13 (17.3) |
| Stomach aches | 1 | 1 (1.3) | 0 | 0 (0.0) | 2 | 2 (2.7) | 0 | 0 (0.0) |
| Lack of appetite | 0 | 0 (0.0) | 1 | 1 (1.3) | 0 | 0 (0.0) | 1 | 1 (1.3) |
| Sleep problems | 0 | 0 (0.0) | 17 | 15 (20.0) | 0 | 0 (0.0) | 8 | 7 (9.3) |
| Frustration | 0 | 0 (0.0) | 3 | 3 (4.0) | 0 | 0 (0.0) | 1 | 1 (1.3) |
| Lack of confidence | 0 | 0 (0.0) | 0 | 0 (0.0) | 0 | 0 (0.0) | 0 | 0 (0.0) |
| Distractibility | 0 | 0 (0.0) | 0 | 0 (0.0) | 0 | 0 (0.0) | 0 | 0 (0.0) |
| Other physiological event | 21 | 19 (25.3) | 12 | 10 (13.3) | 21 | 19 (25.3) | 6 | 4 (5.3) |
| Other psychological event | 3 | 2 (2.7) | 7 | 5 (6.7) | 2 | 2 (2.7) | 3 | 3 (4.0) |

**Note. AE** = Adverse event; AES=Adverse Events; ADES=Adverse Device Effects.

**Extended Data Table 6 | Categorisation of adverse events severity**

| AE Category | Real TNS | | Sham TNS | |
|---|---|---|---|---|
| | N events | N with event (%) | N events | N with event (%) |
| Mild | 95 | 52 (69.3) | 78 | 46 (61.3) |
| Moderate | 2 | 2 (2.7) | 6 | 5 (6.7) |
| Severe | 0 | 0 (0.0 | 0 | 0 (0.0) |

**Extended Data Table 7 | Acceptability questionnaire completed by parents and children at week 4**

| Acceptability Questionnaire at week 4 | Parent | | Child | |
|---|---|---|---|---|
| | Real TNS | Sham TNS | Real TNS | Sham TNS |
| How would you rate your child's side effects? | | | | |
| No particular side effects | 55 (75.3) | 66 (89.2) | 48 (65.8) | 55 (74.3) |
| Mild side effects | 15 (20.5) | 7 (9.5) | 21 (28.8) | 12 (16.2) |
| Moderate side effects | 2 (2.7) | 0 (0.0) | 4 (5.5) | 6 (8.1) |
| Severe side effects | 1 (1.4) | 1 (1.4) | 0 (0.0) | 1 (1.4) |
| Extremely severe side effects | 0 (0.0) | 0 (0.0) | 0 (0.0) | 0 (0.0) |
| How would you rate the burden of doing TNS every night for four weeks? | | | | |
| 0. No particular burden/ bother | 33 (45.2) | 42 (56.8) | 22 (30.1) | 34 (45.9) |
| 1. Mild burden/ bother | 31 (42.5) | 26 (35.1) | 34 (46.6) | 31 (41.9) |
| 2. Moderate burden/ bother | 6 (8.2) | 6 (8.1) | 12 (16.4) | 7 (9.5) |
| 3. Severe burden/ bother | 2 (2.7) | 0 (0.0) | 3 (4.1) | 2 (2.7) |
| 4. Extremely severe burden/bother | 1 (1.4) | 0 (0.0) | 2 (2.7) | 0 (0.0) |
| Do you think the treatment your child received would be an acceptable treatment to give to other people with ADHD? | | | | |
| No | 3 (4.1) | 2 (2.7) | 5 (6.8) | 0 (0.0) |
| Not sure | 8 (11.0) | 12 (16.2) | 10 (13.7) | 14 (18.9) |
| Maybe | 16 (21.9) | 11 (14.9) | 26 (35.6) | 26 (35.1) |
| Yes | 23 (31.5) | 28 (37.8) | 25 (34.2) | 22 (29.7) |
| Definitely yes | 23 (31.5) | 21 (28.4) | 7 (9.6) | 12 (16.2) |

**Extended Data Table 8 | Complier Average Causal Effect Analysis (CACE) of primary outcome ADHD-RS total score at week 4**

| Analysis | Mean difference (95% CI) | Cohen's d (95% CI) | p-value |
|---|---|---|---|
| ADHD-RS at Week 4 – CACE estimate - average treatment effect in those who complied with intervention (n=150) | 1.12 (-1.38, 3.61) | 0.12 (-0.14, 0.37) | 0.381 |
| ITT analysis of ADHD-RS at Week 4 (for reference) | 0.83 (-2.47, 4.13) | 0.09 (-0.26, 0.43) | 0.622 |

**Note** ITT=Intention to Treat; CACE=Complier Average Causal Effect; CI=Confidence Interval. P-values were calculated using two-sided z-tests from the linear mixed models as outlined in methods.

**Extended Data Table 9 | Subgroup analysis of primary outcome ADHD-RS total score at week 4 in participants who were off medication/medication-naïve at baseline**

| Analysis | Mean difference (95% CI) | Cohen's d (95% CI) | p-value |
|---|---|---|---|
| ADHD-RS at Week 4 in medication naive or off stimulant medication at baseline (n=91) | 0.68 (-3.59, 4.95) | 0.07 (-0.37,0.51) | 0.755 |
| ADHD-RS at Week 4 in those on stimulant medication at baseline (n=59) | 1.06 (-4.19, 6.32) | 0.11 (-0.43,0.65) | 0.691 |
| ITT analysis of ADHD-RS at Week 4 (for reference) | 0.83 (-2.47,4.13) | 0.09 (-0.26,0.43) | 0.622 |

**Note**. ITT=Intention to Treat. P-values were calculated using two-sided z-tests from the linear mixed models as outlined in methods.

# Reporting Summary

## Statistics

For all statistical analyses, confirm that the following items are present in the figure legend, table legend, main text, or Methods section.

| n/a | Confirmed | |
|---|---|---|
| ☐ | ☒ | The exact sample size (*n*) for each experimental group/condition, given as a discrete number and unit of measurement |
| ☐ | ☒ | A statement on whether measurements were taken from distinct samples or whether the same sample was measured repeatedly |
| ☐ | ☒ | The statistical test(s) used AND whether they are one- or two-sided *Only common tests should be described solely by name; describe more complex techniques in the Methods section.* |
| ☐ | ☒ | A description of all covariates tested |
| ☐ | ☒ | A description of any assumptions or corrections, such as tests of normality and adjustment for multiple comparisons |
| ☐ | ☒ | A full description of the statistical parameters including central tendency (e.g. means) or other basic estimates (e.g. regression coefficient) AND variation (e.g. standard deviation) or associated estimates of uncertainty (e.g. confidence intervals) |
| ☐ | ☒ | For null hypothesis testing, the test statistic (e.g. *F*, *t*, *r*) with confidence intervals, effect sizes, degrees of freedom and *P* value noted *Give P values as exact values whenever suitable.* |
| ☒ | ☐ | For Bayesian analysis, information on the choice of priors and Markov chain Monte Carlo settings |
| ☐ | ☒ | For hierarchical and complex designs, identification of the appropriate level for tests and full reporting of outcomes |
| ☐ | ☒ | Estimates of effect sizes (e.g. Cohen's *d*, Pearson's *r*), indicating how they were calculated |

*Our web collection on statistics for biologists contains articles on many of the points above.*

## Software and code

Policy information about availability of computer code

| | |
|---|---|
| Data collection | the MACRO electronic data capture (EDC) system (v4.15.0.116) was used to store clinical trial data. The Tobii Pro Eye Tracker Manager software (Tobii AB, Stockholm, Sweden,Tobii Pro Lab v1.207 ) was utilized to process pupillometry data. Computerized cognitive task data were processed through Microsoft EXCEL. Objective Hyperactivity data were processed through the Empatica E4 Manager software (Empatica Srl, Milan, Italy v2.0.3 (5119)). |
| Data analysis | Data analysis was conducted using Stata 18 (StataCorp LLC, v18.0) in accordance with a pre-specified statistical analysis plan. No custom or unpublished software was used. |

For manuscripts utilizing custom algorithms or software that are central to the research but not yet described in published literature, software must be made available to editors and reviewers. We strongly encourage code deposition in a community repository (e.g. GitHub). See the Nature Portfolio guidelines for submitting code & software for further information.

# Data

Policy information about availability of data

All manuscripts must include a data availability statement. This statement should provide the following information, where applicable:
- Accession codes, unique identifiers, or web links for publicly available datasets
- A description of any restrictions on data availability
- For clinical datasets or third party data, please ensure that the statement adheres to our policy

All data supporting the findings of this study have been deposited in the Figshare repository and are publicly available at https://doi.org/10.6084/m9.figshare.29414744.v1. Source data underlying the figures and tables presented in this paper are included in the Figshare repository. No custom code was generated or used in this study.

# Research involving human participants, their data, or biological material

Policy information about studies with human participants or human data. See also policy information about sex, gender (identity/presentation), and sexual orientation and race, ethnicity and racism.

| | |
|---|---|
| Reporting on sex and gender | Biological sex assigned at birth (male or female) was collected as reported by parents/guardians during eligibility assessments. Sex at birth was considered in the study design and was used both as a stratification factor during randomization and as a covariate in the primary and secondary statistical analyses. Overall, 97 male and 53 female participants were recruited. In the real TNS arm, 65.3% of participants were male and 34.7% were female, while in the sham TNS arm, 64.0% were male and 36.0% were female. In accordance with the Sex and Gender Equity in Research (SAGER) guidelines, we conducted post hoc analyses of the primary outcome disaggregated by sex at birth. |
| Reporting on race, ethnicity, or other socially relevant groupings | Ethnicity was self-reported by participants' parents or guardians at eligibility using standardized UK Census categories (e.g., White, Black/African/Caribbean/Black British, Asian/Asian British, Mixed/Multiple ethnic groups, and Other). The purpose of collecting this data was to assess the representativeness of the study sample. This variable was not used as a proxy for socioeconomic status, which was independently assessed using the Index of Multiple Deprivation. The sample was broadly representative of the UK general population, with 79.3% identifying as White and 20.7% from other ethnic groups. To control for potential confounders, statistical analyses employed longitudinal linear mixed models with fixed effects for key covariates, including age group (8–13.5 years vs. 13.6–19 years), sex, site (King's College London vs. University of Southampton), baseline ADHD symptom severity (ADHD-RS score), and medication status (on stable stimulant medication vs. off/medication-naïve). These adjustments were specified a priori and implemented consistently across outcome models to reduce bias and improve the precision of treatment effect estimates. |
| Population characteristics | The study population consisted of 150 children and adolescents aged 8 to 18 years (mean age = 12.6 years, SD = 2.8) who met the diagnostic criteria for Attention-Deficit/Hyperactivity Disorder (ADHD) according to the DSM-5, as assessed using the semi-structured Kiddie Schedule for Affective Disorders and Schizophrenia (K-SADS). The sample included 64.7% males and 35.3% females. In terms of ADHD subtypes, 88.7% had the combined presentation, 10.7% had the inattentive presentation, and 0.7% had the hyperactive/impulsive presentation. Ethnically, the sample was 79.3% White, 10% Mixed or Multiple ethnic groups, 4.7% Asian or Asian British, 3.3% Black, African, Caribbean, or Black British, and 2.7% from other ethnic backgrounds. This distribution closely mirrors the general UK population, enhancing generalizability. Participants' IQ was within the typical range, with a mean Full Scale IQ (WASI FSIQ-4) of 107.6 (SD = 13.8). Comorbid Oppositional Defiant Disorder was present in 36% of participants, and Conduct Disorder in 2.7%. At baseline, 60.7% of participants were medication-naïve or off medication, and 39.3% were on stable stimulant medication. Participants were stratified at randomization by age (8–13.5 vs. 13.6–19 years), sex, site (King's College London or University of Southampton), and medication status. The cohort represented a diverse range of socioeconomic backgrounds as assessed by the Index of Multiple Deprivation (mean = 6.7, SD = 2.7). |
| Recruitment | Participants were recruited between September 2022 and November 2024 from public and private clinics in Greater London, Southampton, and Portsmouth; nationwide parent and ADHD support groups; general practitioners (GPs); the NHS Consent for Contact (C4C) research directory; and social media. To minimize selection bias and enhance external validity, recruitment efforts targeted both urban and suburban areas in London and Southampton and sought to include families from a range of socioeconomic backgrounds. Stratified randomization was employed to ensure balance across key variables (age, sex, site, and medication status). The use of broad eligibility criteria and the inclusion of both medicated and non-medicated participants further reduced the risk of sampling bias. To support engagement, participation incentives and reimbursement of travel expenses were provided. Children and their parents/carers gave both digital and written informed consent/assent. Participants were reimbursed for travel costs and received up to £350 for study completion (or up to £450 for those enrolled in the fMRI sub-study, which will be published separately). |
| Ethics oversight | The trial was approved by the West Midlands–Solihull NHS Research Ethics Committee (REC; Ref: 21/WN/0169) and the Medicines and Healthcare products Regulatory Agency (MHRA; Ref: CI/2022/0003/GB). It was conducted in accordance with the Declaration of Helsinki 1975 and is reported following CONSORT guidelines. Independent oversight of the trial was provided by a data monitoring committee and a trial steering committee. |

Note that full information on the approval of the study protocol must also be provided in the manuscript.

# Field-specific reporting

Please select the one below that is the best fit for your research. If you are not sure, read the appropriate sections before making your selection.

☒ Life sciences ☐ Behavioural & social sciences ☐ Ecological, evolutionary & environmental sciences

For a reference copy of the document with all sections, see nature.com/documents/nr-reporting-summary-flat.pdf

# Life sciences study design

All studies must disclose on these points even when the disclosure is negative.

| | |
|---|---|
| Sample size | The estimated sample size of 128 participants (64:64) was calculated using a baseline to post-treatment correlation of 0.5, 90% power, 5% type I error, and an anticipated effect size of 0.5 for a reduction in ADHD symptoms at 4 weeks. The number of participants was inflated to 150 (75:75) to account for a loss to follow-up rate of 15%. |
| Data exclusions | Data were analyzed using an intention-to-treat (ITT) approach, with all randomized participants included in the final analyses. |
| Replication | This study was a multi-centre, double-blind, randomized, sham-controlled, parallel-group, phase IIb RCT investigating the efficacy and safety of external trigeminal nerve stimulation (TNS) on core symptoms of ADHD and related clinical, cognitive, and physiological outcomes in youths with ADHD. It was conducted across two independent sites in the UK (King's College London and the University of Southampton), and It was designed as a confirmatory study following a previous pilot RCT conducted by another research group in the US (McGough et al., 2019). No experimental conditions were repeated in independent cohorts beyond this trial.<br>Reference:<br>McGough, J. J., Sturm, A., Cowen, J., Tung, K., Salgari, G. C., Leuchter, A. F., ... & Loo, S. K. (2019). Double-blind, sham-controlled, pilot study of trigeminal nerve stimulation for attention-deficit/hyperactivity disorder. Journal of the American Academy of Child & Adolescent Psychiatry, 58(4), 403-411.https://doi.org/10.1016/j.jaac.2018.11.013 |
| Randomization | Randomization was done by minimization by sex (male/female), medication status (on medication; off medication/naïve), site (London, Southampton) and age (8–13.5 years; 13.6-19 years) using a validated, online, web-based system from King's Clinical Trials unit (KCTU). |
| Blinding | Participants, parents/carers, postdoctoral research associates, Principal Investigator, Co-Investigators, and analysts were blinded to treatment arm except for the trial manager and trial manager assistants, who trained participants/parents on the device use, but did not conduct research assessments and were prohibited from sharing the information with other team members. Analysts were blinded until after database lock. Blinding was assessed by a questionnaire administered to participants, parents/carers, and researchers after 1 and 4 weeks of TNS treatment. |

# Reporting for specific materials, systems and methods

We require information from authors about some types of materials, experimental systems and methods used in many studies. Here, indicate whether each material, system or method listed is relevant to your study. If you are not sure if a list item applies to your research, read the appropriate section before selecting a response.

## Materials & experimental systems

| n/a | Involved in the study |
|---|---|
| ☒ | ☐ Antibodies |
| ☒ | ☐ Eukaryotic cell lines |
| ☒ | ☐ Palaeontology and archaeology |
| ☒ | ☐ Animals and other organisms |
| ☐ | ☒ Clinical data |
| ☒ | ☐ Dual use research of concern |
| ☒ | ☐ Plants |

## Methods

| n/a | Involved in the study |
|---|---|
| ☒ | ☐ ChIP-seq |
| ☒ | ☐ Flow cytometry |
| ☒ | ☐ MRI-based neuroimaging |

# Clinical data

Policy information about clinical studies

All manuscripts should comply with the ICMJE guidelines for publication of clinical research and a completed CONSORT checklist must be included with all submissions.

| | |
|---|---|
| Clinical trial registration | Trial registration: ISRCTN82129325 |
| Study protocol | The full trial protocol and the SAP has been published. See reference below. -Rubia, K., Johansson, L., Carter, B. et al. The efficacy of real versus sham external Trigeminal Nerve Stimulation (eTNS) in youth with<br>Attention-Deficit/Hyperactivity Disorder (ADHD) over 4 weeks: a protocol for a multi-centre, double-blind, randomized, parallelgroup, phase IIb study (ATTENS). BMC Psychiatry 24, 326 (2024). https://doi.org/10.1186/s12888-024-05650-1 |

| Data collection | Participant recruitment began in September 2022 and concluded in November 2024. Data were collected between September 2022 and March 2025 (6 months follow-up data collection) at two sites in the United Kingdom: King's College London (Institute of Psychiatry, Psychology and Neuroscience) and the University of Southampton (Centre for Innovation in Mental Health). Study assessments were conducted in research facilities within these universities, including laboratories for cognitive testing, physiological measurements, and behavioral assessments, to ensure standardized and controlled data collection. |
|---|---|
| Outcomes | The primary outcome measure was the investigator-scored, parent-rated ADHD-RS total score collected at eligibility, baseline and weekly throughout the four-week trial. Secondary outcome measures were collected at baseline, week 4, and at 6 months follow-up and included the following rating scales: teacher-rated ADHD-RS (school version) , Conners Teacher Rating Scale short form T-S , child-reported Strength and Difficulties Questionnaire (SDQ) , parent and child-reported Affective Reactivity Index (ARI) , parent and child-reported Child and Adolescent Anxiety and Depression scale (RCADS-25), child-reported Columbia Suicide Severity Rating Scale (C-SSRS) , child-reported Mind Excessively Wandering Scale (MEWS), parent-reported Sleep Disturbance Scale for Children (SDSC), and the investigator scored parent-rated ADHD-RS at 6 months follow-up. Vigilance (omission and commission errors) was assessed using the Mackworth Clock Task. Pupillometry data were recorded with the Tobii Pro Nano screen-based eye-tracking device (Tobii AB, Stockholm, Sweden) during a 1-minute resting condition and a cognitive task. Objective hyperactivity, defined as the composite score of both the intensity (g) and frequency (g) of movement, was assessed at baseline and week 4 using a 3-axis accelerometer embedded in the Empatica E4 wristband device (Empatica Srl, Milan, Italy). Other measures included an acceptability questionnaire filled out by participants and their parents/carers at the end of the treatment, side effects questionnaires and open-ended adverse event forms completed by participants and their parents/carers at baseline, week 4, and at 6 months follow-up. Blinding was assessed by a questionnaire administered to participants, parents/carers, and researchers after 1 and 4 weeks of TNS treatment. |

# Plants

| Seed stocks | *Report on the source of all seed stocks or other plant material used. If applicable, state the seed stock centre and catalogue number. If plant specimens were collected from the field, describe the collection location, date and sampling procedures.* |
|---|---|
| Novel plant genotypes | *Describe the methods by which all novel plant genotypes were produced. This includes those generated by transgenic approaches, gene editing, chemical/radiation-based mutagenesis and hybridization. For transgenic lines, describe the transformation method, the number of independent lines analyzed and the generation upon which experiments were performed. For gene-edited lines, describe the editor used, the endogenous sequence targeted for editing, the targeting guide RNA sequence (if applicable) and how the editor was applied.* |
| Authentication | *Describe any authentication procedures for each seed stock used or novel genotype generated. Describe any experiments used to assess the effect of a mutation and, where applicable, how potential secondary effects (e.g. second site T-DNA insertions, mosiacism, off-target gene editing) were examined.* |

