## [Peer Review File · Nature Medicine]

External Trigeminal Nerve Stimulation in youth with ADHD: a randomized, sham-controlled, phase 2b trial

Corresponding Author: Professor Katya Rubia

Version 0:

Reviewer comments:

Reviewer #1

(Remarks to the Author)

This study is a multicenter, double-blind, randomized, sham-controlled, parallel-group phase 2b trial investigating the short- and long-term efficacy of TMS in 150 patients recruited across two centers in the United Kingdom with ADHD. A notable innovation of this trial is the inclusion of a sham TMS condition, which delivered a 30-second stimulation per hour at a lower frequency. Overall, the results showed no differential treatment effects on the primary outcome and nearly all secondary outcomes, except for a modest effect on mind-wandering favoring the real TMS group. Adherence was excellent, and blinding was very successful. The findings also contribute to the literature supporting the safety of neuromodulation techniques, although efficacy was limited in this. The manuscript is very well written, with clear presentation of statistical analyses and results. I have only a few minor comments:

1. Mind-Wandering and Vigilance Outcomes

Unlike outcomes such as cognitive functioning, depression, anxiety, emotional dysregulation, and sleep—which are familiar to most clinicians—mind-wandering and vigilance are less commonly used in clinical practice. Briefly describing their relevance to ADHD would aid reader understanding.

2. Flanker Task and Cognitive Reaction Time

In the Discussion section (around line 353), you note a significant reduction in flanker task incongruent reaction time after 8 weeks of treatment. Further explanation of what “flanker task incongruent reaction time” will be helpful.

3. Mechanism of Exclusion Criteria

In the Methods, you note that atomoxetine, and guanfacine were excluded because they may interfere with the TMS mechanism. It would strengthen the manuscript to briefly explain the rationale—i.e., what mechanistic overlap or interaction was anticipated between these medications and TMS.

Overall, the authors have done a tremendous job in conducting this rigorously controlled trial with improved control conditions relative to prior studies.

[Name Redacted]

Reviewer #2

(Remarks to the Author)

The manuscript describes the design clearly with a detailed flowchart of the recruitment. The primary and secondary outcome measures were clearly defined, and the results were presented in an appropriate order along with the confidence intervals.

Comments:

(1) Due to the potential effects of low-frequency stimulation of the Sham TNS, it will be helpful to add another control group in which participants wear the same disposable self-adhesive patch electrodes, but no stimulation is delivered.

(2) The age range varies at different sections of the manuscript, and it is not clear whether 19 years old is included or not:

- In Table 1. The age range is 8-19.

- In the “Randomization and blinding” section on page 18, it says “and age (8-13.5 years; 13.6-19 years);

- In the “Participants” section on page 18, it says, “One hundred and fifty children and adolescents (8-18 years) with ...”

(3) There is a lack of a detailed description of the statistical analysis, and it would be helpful to specify the statistical model

with all variables, along with the assumptions of random effects.

(4) It is unclear how the covariates were selected for the model and how each variable was coded. For example, it is not clear how age was included in the model with the two age groups. Given the wide range of ages (8-18 yrs), it would be interesting to check if age is significant.

Reviewer #3

(Remarks to the Author)

The authors explored the efficacy of TNS in children and adolescents with ADHD in a multi-centre, double-blind, randomized, sham-controlled, parallel-group clinical trial. In summary, there were no observed significant differences on the primary outcome. Among a broad number of secondary outcomes, including clinical, cognitive, and physiological outcomes, only mind wandering showed a small reduction favoring real TNS, which may simply reflect the large number of comparisons performed. The authors additionally performed a pre-specified subgroup analysis on participants who were not on stimulant treatment, which was also negative.

This study is highly relevant, especially in light of the FDA's 2019 clearance of TNS as the first non-pharmacological treatment for ADHD. Notably, the authors' improved sham design appears to have increased blinding success, suggesting that earlier positive trials using less rigorous shams may have overestimated the efficacy. Overall, this is a very well-designed randomized trial with an a priori-published protocol, which enhances its credibility.

I have a few suggestions and questions:

1. You report the percentage of participants on stimulants in each group, but it would be helpful to specify which stimulant agents as well as the dose ranges used.
2. The placebo effect observed in this study appears substantial. Although this has been discussed already, I suggest the authors explicitly describe how it compares with placebo responses reported in other clinical trials in children with ADHD.
3. The Methods indicate nightly sleep diaries were used to record hours of device use. Was there any objective device-logged data to corroborate the self-report? If not, please discuss the limitations of this approach.
4. Aside from tracking total wear time, did the study include any technical verification (e.g., impedance monitoring or device-logged usage data) to ensure continuous stimulation each night? If not, please acknowledge this as a potential limitation.

Negative clinical trials typically receive less media attention and are, on average, less likely to be published in high-impact journals. However, I believe this well-designed study will help reverse that bias, meaningfully advance the field, and deserves appropriate recognition.

[Name Redacted]

Version 1:

Reviewer comments:

Reviewer #1

(Remarks to the Author)

The authors have satisfactorily addressed all questions and concerns raised during the initial review. No further comments.

Reviewer #2

(Remarks to the Author)

Thank you for the thoughtful responses.

Reviewer #3

(Remarks to the Author)

I have no further questions, and I thank the authors for appropriately addressing my previous concerns.

[Name Redacted]

Department of Child &
Adolescent Psychiatry
Institute of Psychiatry,
Psychology and Neuroscience

Katya Rubia, PhD
Professor of
Cognitive
Neuroscience;
Deputy Head of
Department

PO85
16 De Crespigny Park
London SE5 8AF
Telephone 020 7848
0463
Email:
katya.rubia@kcl.ac.uk

London, 21th of July, 2025

Dear Prof Staal

We thank you very much for sending us the comments from the reviewers and the Editor for our article "The efficacy of external Trigeminal Nerve Stimulation (TNS) in youth with Attention-Deficit/Hyperactivity Disorder (ADHD): a multi-centre, double-blind, randomized, sham-controlled, parallel-group, phase IIb trial" (ref: NMED-142790). We found the comments very helpful and think they have substantially improved our MS. We have now addressed all comments and hope the MS is now ready for acceptance in Nature Medicine.

Katya Rubia and Aldo Conti

Below you find the point-by-point response to the Referees' and the Editor's comments:

Response to Referees

Reviewer #1

This study is a multicenter, double-blind, randomized, sham-controlled, parallel-group phase 2b trial investigating the short- and long-term efficacy of TMS in 150 patients recruited across two centers in the United Kingdom with ADHD. A notable innovation of this trial is the inclusion of a sham TMS condition, which delivered a 30-second stimulation per hour at a lower frequency. Overall, the results showed no differential treatment effects on the primary outcome and nearly all secondary outcomes, except for a modest effect on mind-wandering favoring the real TMS group. Adherence was excellent, and blinding was very successful. The findings also contribute to the literature supporting the safety of neuromodulation techniques, although efficacy was limited in this. The manuscript is very well written, with clear presentation of statistical analyses and results.

Response: *We thank [Name Redacted] for [the] positive comments.*

I have only a few minor comments:

1. Mind-Wandering and Vigilance Outcomes

Unlike outcomes such as cognitive functioning, depression, anxiety, emotional dysregulation, and sleep—which are familiar to most clinicians—mind-wandering and vigilance are less commonly used in clinical practice. Briefly describing their relevance to ADHD would aid reader understanding.

Response: Thank you for this fair comment. We have now expanded on both issues of Mindwandering and Vigilance to make the clinical relevance clearer to the reader's understanding.

Vigilance:

We have now added a sentence in the introduction on page 3 on impairments in ADHD in executive functions and in tasks of attention and vigilance.

“ADHD is also associated with impairments in executive functions, including in tasks of sustained attention and vigilance^{3,4}”

We also added a sentence on how ADHD medications improve executive functions and vigilance (P.3)

“Both stimulants and non-stimulants have shown to also improve performance in executive function tasks including sustained attention and vigilance in children and adults with ADHD¹¹”

Mind-wandering:

We now added a sentence at the end of the introduction on page 4 on the fact that mind-wandering is impaired in ADHD.

“There is consistent evidence that children with ADHD have increased mind-wandering which interferes with their cognitive performance, in particular in tasks of sustained attention and vigilance²⁷”

We also added a sentence in the discussion on page 11.

“Mind-wandering has been found to be a key behavioural impairment in people with ADHD which is thought to interfere with cognitive/attention performance²⁷. This is further underpinned by consistent evidence at the brain level for increased activation in people with ADHD of the default mode network – which mediates mind-wandering – during cognitive and attention task performance and during rest^{23,24,34,35} and by evidence for a poor anti-correlation between the default mode network and attention networks in people with ADHD relative to healthy controls³⁵.”

2. Flanker Task and Cognitive Reaction Time

In the Discussion section (around line 353), you note a significant reduction in flanker task incongruent reaction time after 8 weeks of treatment. Further explanation of what “flanker task incongruent reaction time” will be helpful.

Response: This sentence has now been reworded (P. 11):

“While an open-label pilot study of TNS reported a significant reduction in flanker task incongruent reaction times to incongruent trials in the flanker task (i.e., reaction times to incongruent trials are typically slower than those to congruent trials, which is an indicator of interference inhibition) after eight weeks of treatment²⁶, this finding was not replicated in the subsequent double-blind pilot RCT²⁰”

3. Mechanism of Exclusion Criteria

In the Methods, you note that atomoxetine, and guanfacine were excluded because they may interfere with the TMS mechanism. It would strengthen the manuscript to briefly explain the rationale—i.e., what mechanistic overlap or interaction was anticipated between these medications and TNS.

Response: Thank you for this helpful comment. We have now briefly explained the rationale in the methods section on page 23. See also below:

“Participants were also excluded if they were medicated with non-stimulants such as atomoxetine, guanfacine or clonidine. Non-stimulant medications have shown to enhance noradrenaline in frontal and cortical regions via selectively blocking noradrenaline transporters (Atomoxetine) or by stimulating postsynaptic α_2 -adrenergic receptors (Guanfacine and Clonidine) ⁵². Given that a key mechanism of action of TNS is thought to be the stimulation of the locus coeruleus which releases noradrenaline into the brain¹⁵, we excluded these medications due to their similar underlying mechanisms of action to TNS^{14,16} and potential interaction effects.”

Reviewer #2 (Remarks to the Author):

The manuscript describes the design clearly with a detailed flowchart of the recruitment. The primary and secondary outcome measures were clearly defined, and the results were presented in an appropriate order along with the confidence intervals.

Response: We thank the reviewer for the positive comments.

Comments:

(1) Due to the potential effects of low-frequency stimulation of the Sham TNS, it will be helpful to add another control group in which participants wear the same disposable self-adhesive patch electrodes, but no stimulation is delivered.

Response: The study design and protocol were prespecified and we will not be able to add another control condition at this stage. Also, the rigorous sham condition—as also noted by Referee 1 and Referee 3—was a key “innovation”, “an improved sham design” which “has improved blinding success” and has “substantial implications on earlier sham designs that have overestimated the efficacy”. We therefore do not think this is a weakness but a fundamental strength of the study design. Adding a no stimulation group in future studies would compromise blinding and therefore result in a placebo confound.

(2) The age range varies at different sections of the manuscript, and it is not clear whether 19 years old is included or not:

- In Table 1. The age range is 8-19.
- In the “Randomization and blinding” section on page 18, it says “and age (8-13.5 years; 13.6-19 years);

• In the “Participants” section on page 18, it says, “One hundred and fifty children and adolescents (8-18 years) with ...”

Response: *We apologise for the confusion. The inclusion criteria for the age range were 8-18 years at the consent stage. Four children were 18 at consent stage but turned 19 before randomisation took place. We have now clarified this in the participants section (P.5).*

“While the inclusion criteria for the age range was 8-18 years at the consent stage, four children turned 19 before randomisation took place.”

(3) There is a lack of a detailed description of the statistical analysis, and it would be helpful to specify the statistical model with all variables, along with the assumptions of random effects.

Response: *Thank you for this comment. We have now made the statistical analysis clearer in the main text (P. 28), including the assumed covariance structure (independent) for the random effects. Further details on the statistical analysis methods are given in the supplementary material.*

*“For the primary analysis, a longitudinal linear mixed model was used, fitting 4-week ADHD symptom scores as a continuous outcome, with continuous time as a covariate using actual **observed** time of assessments and an interaction between time and trial arm to estimate **treatment** effects at week 1, 2,3 and 4 using post-estimation. A random intercept was included as well as a random slope over time for each participant, **and assuming an independent covariance matrix for these random effects.**”*

(4) It is unclear how the covariates were selected for the model and how each variable was coded. For example, it is not clear how age was included in the model with the two age groups. Given the wide range of ages (8-18 yrs), it would be interesting to check if age is significant.

Response: *Covariates in the model were pre-specified in the SAP and based on clinical knowledge i.e. these covariates are known prognostic factors for the ADHD-RS outcome and were also therefore included as stratifiers in the randomisation (we have therefore “analysed as we randomised” as is best practice). Age was included as a binary factor, as were the other stratifiers used as covariates (site, medication and sex at birth) which we have made clearer in the main text (P.28). We did carry out a post-hoc subgroup analysis of the effect in a younger age range (8-12) that did not show a statistically significant treatment effect on the ADHD-RS outcome as reported in the results section.*

*“We additionally adjusted for fixed effects of baseline ADHD-RS score, site (**London, Southampton**), age **category** (8– 13.5 years; 13.6-19 years), **sex at birth** (**male, female**), and medication status (**On stable medication, Off medication/naive**). An adjusted mean difference (aMD) was calculated between the treatment groups with associated 95% confidence intervals (95% CI) and p-value (for week 4 only).”*

Reviewer #3 (Remarks to the Author):

The authors explored the efficacy of TNS in children and adolescents with ADHD in a multi-centre, double-blind, randomized, sham-controlled, parallel-group clinical trial. In summary, there were no observed significant differences on the primary outcome. Among a broad number of secondary outcomes, including clinical, cognitive, and physiological outcomes, only mind wandering showed a small reduction favoring real TNS, which may simply reflect the large number of comparisons performed. The authors additionally performed a pre-specified subgroup analysis on participants who were not on stimulant treatment, which was also negative.

This study is highly relevant, especially in light of the FDA's 2019 clearance of TNS as the first non-pharmacological treatment for ADHD. Notably, the authors' improved sham design appears to have increased blinding success, suggesting that earlier positive trials using less rigorous shams may have overestimated the efficacy. Overall, this is a very well-designed randomized trial with an a priori-published protocol, which enhances its credibility.

Response: We thank [Name Redacted] for [the] positive comments.

I have a few suggestions and questions:

1. You report the percentage of participants on stimulants in each group, but it would be helpful to specify which stimulant agents as well as the dose ranges used.

Response: We thank the reviewer for this insightful comment which allowed us to add a table in the supplement with further details on the medications used in both groups. We have now specified the stimulant medications and the mean dose/dose ranges for the real TNS group and sham TNS group in Table S1 (see also below) that is now referenced in the main text, on page 5, and included in the supplementary material as Table S1. Of note, in the UK, methylphenidate is by far the most commonly used stimulant in clinical practice.

Table S1. Stimulant medications, mean daily dose, and dose ranges recorded at baseline.

Medication at baseline	Methylphenidate		Amphetamines	
	N (%)	Mean daily dose (SD) (mg) [range]	N (%)	Mean daily dose (SD) (mg) [range]
Real TNS (N=75)	26 (34.7%)	22.8 (15.9) [5-90]	3 (4.0%)	35.0 (12.9) [20-50]
Sham TNS (N=75)	22 (29.3%)	30.2 (17.3) [5-72]	8 (10.7%)	33.2 (17.6) [5-60]

2. The placebo effect observed in this study appears substantial. Although this has been discussed already, I suggest the authors explicitly describe how it compares with placebo responses reported in other clinical trials in children with ADHD.

Response: We thank [Name Redacted] for this very helpful comment and are grateful for the opportunity to compare our placebo response with the placebo responses typically found in the literature of clinical medication trials in ADHD.

As can be observed in Table 1, at primary endpoint of 4 weeks, the placebo group improved by 10 scores on the ADHD-RS, which is equivalent to an effect size of Cohen's *d* of 0.9. The pooled effect size for the placebo response based on parent-rated ADHD severity scores in a meta-analysis of placebo responses in 27 RCTs of placebo controlled medication studies shows a Cohen's *d* of 0.4 for parent ratings of the ADHD-RS. This confirms that the placebo response is larger for a neurotechnology-based intervention like TNS, as already suggested by the literature. We have now added a sentence about this in the discussion on page 10:

*"In fact, the sham group improved by 10 points on the parent-rated ADHD-RS, which is equivalent to a large Cohen's *d* of 0.9, which is more than double the pooled medium effect size for a placebo effect of 0.4 for parent ratings of the ADHD-RS reported in a meta-analysis of 27 RCTs of medication and placebo effects in ADHD³³. Our findings hence extend previous evidence in the literature^{29,30} that the placebo response related to a neurotechnology such as TNS is larger than the typical placebo response in medication trials."*

3. The Methods indicate nightly sleep diaries were used to record hours of device use. Was there any objective device-logged data to corroborate the self-report? If not, please discuss the limitations of this approach.

Response: Thank you for raising this important point. The study relied on self-reported nightly sleep diaries to record hours of device use and nightly setting. We aimed to include an objective measure of compliance. Unfortunately, the objective measures of the Monarch NeuroSigma device were not reliable and could not be used. We acknowledge this as a limitation, as self-reported data can be subject to recall bias or inaccuracies. We have now discussed this limitation on P. 12.

"Also, while adherence was very high (94%), it was self-reported and may have been overestimated due to social desirability bias. Adherence relied on participant-completed nightly sleep diaries to track device use. Unfortunately, these could not be corroborated by objective device-logged usage data as they were found not to be reliable. Future studies should incorporate reliable and accurate objective device usage monitoring to improve the accuracy of adherence assessment and ensure treatment fidelity."

4. Aside from tracking total wear time, did the study include any technical verification (e.g., impedance monitoring or device-logged usage data) to ensure continuous stimulation each night? If not, please acknowledge this as a potential limitation.

Response: Please see the response to point 3.

Negative clinical trials typically receive less media attention and are, on average, less likely to be published in high-impact journals. However, I believe this well-designed study will help reverse that bias, meaningfully advance the field, and deserves appropriate recognition.

[Name Redacted]

Response: *we agree.*

Editorial points:

Editorial points to be addressed. Please note that not all of these might apply to your manuscript, in which case please respond with “not applicable.” Structural changes to re-organise the manuscript do not need to be tracked.

* Abstract should be no more than 200 words and for trials should adhere to the CONSORT framework. You must state if the primary outcome was or was not met and provide the effect size and relevant uncertainty estimate. The conclusion of the study must focus only on the primary outcome and safety/tolerability. You must report either all or none of the secondary outcomes, given the space limitation, I would suggest removing all mention of secondary outcomes from the abstract.

Response: *The abstract originally contained 261 words at the time of submission. It has now been revised and reduced to 200 words to meet the journal's word limit.*

* Please include the trial registration number at the end of the Abstract.

Response: *Not Applicable-This was already included.*

* The Introduction should be written for a broad, non-specialist medical reader and provide sufficient context for the work.

Response: *Not Applicable-This was already provided.*

* Please provide details of your cohort in the first paragraph of the Results. This includes number of individuals screened for enrolment, as well as exact dates of first and last patient enrolment. The CONSORT patient disposition diagram and the baseline characteristics must be included as main non-text items (for which there is a strict limit of 6 tables and/or figures).

Response: *Not Applicable-This was already included.*

* The Results should be structured as followed:

- Patient disposition
- Primary outcome(s)
- Secondary outcomes
- Safety
- Exploratory outcomes
- Sensitivity analyses

- Post-hoc analyses

Response: *The results section has been re-structured as specified above.*

* Please remove any subheadings from the Discussion.

Response: *Subheadings have been removed.*

* Please ensure all results are presented in the Results section, no new data should be introduced in the Discussion.

Response: *Not Applicable-No new data are introduced in the discussion.*

* You must include explicit paragraphs of study limitations in the Discussion.

Response: *We have now restructured the study limitations and added a separate paragraph for each of the limitations.*

* The overarching conclusion of the study must be based only on the primary outcome and safety data.

Response: *Not Applicable-This was already the case*

* The Methods should include a full description of the inclusion and exclusion criteria, as well as study procedures and statistical analyses (including a power calculation).

Response: *Not Applicable-This was already included.*

*Please upload the protocol and SAP with the revision materials, so that reviewers and editors have access to them.

Response: *We have uploaded the protocol and SAP at the time of first submission, and we have uploaded it again with the revision material.*

* You must ensure that contributions from all individuals in the author list are available in the Author Contributions statement.

Response: *Not Applicable-This was already ensured.*

* Please move all funding sources to the Acknowledgements, including a statement on the role of the funder.

Response: *All funding sources have been moved to Acknowledgements as well as a statement on the role of the funder.*

* Please ensure that all potential competing interests are detailed for all authors. For any authors with no competing interests, this must also be stated.

Response: *This has been included.*

* Please see our guidelines for the Data and Code Availability Statements. "Available on request" is not acceptable, you must provide details of any restrictions to data and code

availability. <https://www.nature.com/nature-portfolio/editorial-policies/reporting-standards#availability-of-data>

Response: We have provided a new data availability statement on page 29 as data are now publicly available in the Figshare repository. We have also provided a link to the data.

“All data supporting the findings of this study are publicly available in the Figshare repository at <https://doi.org/10.6084/m9.figshare.29414744.v1>”

* The article file must only contain these items in this order:

- Title
- Author List and affiliations
- Abstract
- Introduction
- Results (with Subheadings)
- Discussion
- Acknowledgements
- Author Contributions
- Competing Interests Statement
- References (for main text only)
- Figure legends (for main text only)
- Tables (note: tables should be pasted into Word files as editable tables, not as images)
- Methods
- Data Availability Statement
- Code Availability Statement
- Methods-only References

Response: We have structured the manuscript according to the order of the items specified above.

Other Minor Changes

In addition to the revisions made in response to the reviewers' specific comments, we have made the following minor corrections and clarifications to improve the accuracy and clarity of the manuscript:

- Corrected minor typographical errors throughout the manuscript
- Made minor corrections to supplementary tables and reordered them to align with the structure of the manuscript
- Added Details of Patient Public Involvement (PPI) in design, conduct and reporting of the study on page 23
- Added Trial registration URL and date of registration on page 23
- Updated authors' affiliations